# Iron induces two distinct Ca$^{2+}$ signalling cascades in astrocytes

Wenzheng Guan[1,2,9], Maosheng Xia[1,3,9], Ming Ji[1,4], Beina Chen[1,4], Shuai Li[1,4], Manman Zhang[1,4], Shanshan Liang[1,4], Binjie Chen[1,4], Wenliang Gong[1,4], Chengyi Dong[3], Gehua Wen[1,4], Xiaoni Zhan[1,4], Dianjun Zhang[1,4], Xinyu Li[1,4], Yuefei Zhou[3], Dawei Guan [5✉], Alexei Verkhratsky [4,6,7,8✉] & Baoman Li [1,4✉]

Iron is the fundamental element for numerous physiological functions. Plasmalemmal divalent metal ion transporter 1 (DMT1) is responsible for cellular uptake of ferrous (Fe$^{2+}$), whereas transferrin receptors (TFR) carry transferrin (TF)-bound ferric (Fe$^{3+}$). In this study we performed detailed analysis of the action of Fe ions on cytoplasmic free calcium ion concentration ([Ca$^{2+}$]$_i$) in astrocytes. Administration of Fe$^{2+}$ or Fe$^{3+}$ in μM concentrations evoked [Ca$^{2+}$]$_i$ in astrocytes in vitro and in vivo. Iron ions trigger increase in [Ca$^{2+}$]$_i$ through two distinct molecular cascades. Uptake of Fe$^{2+}$ by DMT1 inhibits astroglial Na$^+$-K$^+$-ATPase, which leads to elevation in cytoplasmic Na$^+$ concentration, thus reversing Na$^+$/Ca$^{2+}$ exchanger and thereby generating Ca$^{2+}$ influx. Uptake of Fe$^{3+}$ by TF-TFR stimulates phospholipase C to produce inositol 1,4,5-trisphosphate (InsP$_3$), thus triggering InsP$_3$ receptor-mediated Ca$^{2+}$ release from endoplasmic reticulum. In summary, these findings reveal the mechanisms of iron-induced astrocytic signalling operational in conditions of iron overload.

[1] Practical Teaching Centre, School of Forensic Medicine, China Medical University, Shenyang, PR China. [2] The First Department of Reproduction, Shengjing Hospital, China Medical University, Shenyang, China. [3] Department of Orthopaedics, The First Hospital, China Medical University, Shenyang, PR China. [4] Department of Forensic Analytical Toxicology, School of Forensic Medicine, China Medical University, Shenyang, China. [5] Department of Forensic Pathology, School of Forensic Medicine, China Medical University, Shenyang, China. [6] Faculty of Biology, Medicine and Health, The University of Manchester, Manchester, UK. [7] Achucarro Center for Neuroscience, IKERBASQUE, Bilbao, Spain. [8] Sechenov First Moscow State Medical University, Moscow, Russia. [9] These authors contributed equally: Wenzheng Guan, Maosheng Xia. ✉email: dwguan@cmu.edu.cn; Alexej.Verkhratsky@manchester.ac.uk; bmli@cmu.edu.cn

Iron contributes to numerous cellular and biochemical processes and acts as a co-factor in various molecular cascades in the nervous tissue including the synthesis and metabolism of several brain-specific enzymes and neurotransmitters[1,2]. In biological systems iron is present in either reduced ferrous ($Fe^{2+}$) or oxidized ferric ($Fe^{3+}$) state. The brain has the second (after liver) highest quantity of iron in the human body with total non-haem iron in the brain reaching about 60 mg[3]. The non-haem iron concentration in the serum ranges between 9 and 30 μM, whereas the iron concentration in cerebrospinal fluid (CSF) is much smaller being around 0.3–0.75 μM[4,5]. Transport of iron across the blood–brain barrier (BBB) is mediated either by transferrin receptor (TFR)-mediated endocytosis of $Fe^{3+}$-bound to transferrin (holo-TF), or, for non-TF-bound iron, by vesicular and non-vesicular pathways[6]. Membrane transport of $Fe^{2+}$ is also mediated by divalent metal ion transporter 1 (DMT1/SLC11A2) which underlies $Fe^{2+}$ uptake through the plasma membrane or from endosomes[6]. Under physiological conditions, the intracellular cytosolic ionised iron levels fluctuate around 0.5–1.5 μM[7].

In the brain, up to three-fourths of total iron is accumulated within neuroglial cells[8]. Astrocytes in particular are fundamental elements of ionostatic control over CNS environment[9]. Ionised $Fe^{2+}$ enters astrocytes through DMT1/SLC11A2 transporters which are particularly concentrated in the endfeet of cerebral and hippocampal astrocytes[10]. In physiological condition, DMT1 is widely distributed in the nervous system, being expressed in neurones[11–13], astrocytes[13–15] and oligodendrocytes[14,16] in vitro and in the brain tissue. In astrocytes, DMT1 mediates non-transferrin-bound iron (NTBI) transport, thus contributing to the brain iron homoeostasis in development and adulthood[17–19]. Conditional deletion of DMT1 form oligodendrocyte precursors substantially inhibited both myelination in development and remyelination in pathology[16]. Evidence for the expression of TFR in astroglial cells remains controversial[6,20,21], while iron overload may influence the expression or distribution of TFR in astrocytic compartments[21,22]. Cellular uptake of $Fe^{3+}$ requires internalization of TF–TFR complex[23]. An adaptor protein Disabled-2 (Dab2) plays an essential role in cell signalling, migration and development[24]. In mice the Dab2 has two isoforms of 96 and 67 kDa (p96 and p67[24]). In human K562 cells, Dab2 regulates internalization of TFR and uptake of TF[25]. Dab2 is also widely distributed in immune cells and in neuroglia[24], although the functional link between Dab-2 and TFR in astrocytes has not been demonstrated.

Astrocytes possess a special form of intracellular ionic excitability, mediated by temporal and spatial fluctuations in the intracellular ion concentration[26,27]. Astroglial $Ca^{2+}$ signalling is mediated by $Ca^{2+}$ release from the endoplasmic reticulum (ER) following activation of inositol-1,4,5-trisphosphate receptor (InsP$_3$R), or intracellular $Ca^{2+}$-gated $Ca^{2+}$ channels known as ryanodine receptors (RyR). Astroglial $Ca^{2+}$ signals may also be generated by plasmalemmal $Ca^{2+}$ entry through $Ca^{2+}$-permeable channels or by sodium-calcium exchanger (NCX) operating in the reverse mode[26,28]. Astroglial $Na^+$ signalling is shaped by plasmalemmal $Na^+$ entry through cationic channels and numerous $Na^+$-dependent transporters, of which the major role belongs to $Na^+$-dependent glutamate transporters[29–31]; $Na^+$ extrusion is mediated by the sodium-potassium pump (NKA). Both $Na^+$ and $Ca^{2+}$ signalling systems are closely coordinated, with NKA and NCX accomplishing this coordination at the molecular level[32]. Astrocytes specifically express α2-subunit containing NKA which is fundamental for astroglial $K^+$ buffering[33]. Astrocytes express all three isoforms of NCX - NCX1/SLC8A1, NCX2/SLC8A2 and NCX3/SLC8A3, with some evidence indicating higher expression of NCX1[34]. The NKA, the NCX and glutamate transporters are known to be preferentially concentrated in the perisynaptic astroglial membranes indicating intimate relationship between these ion-transporting molecules[35,36]. The NCX is also known to localise at caveolae rich in caveolin-3 (Cav-3), the latter isoform being predominantly expressed in astrocytes[37].

In the present paper we performed an in depth analysis of the action of ferrous and ferric ($Fe^{2+}$ and $Fe^{3+}$) on astrocytic $Ca^{2+}$ and $Na^+$ dynamics. We found that $Fe^{2+}$ (through DMT1) and $Fe^{3+}$-TF (through TFR) evoke [$Ca^{2+}$]$_i$ transients in astrocytes in culture and in vivo. Effects of $Fe^{2+}$ on [$Ca^{2+}$]$_i$ are mediated mainly by the reversed NCX, whereas $Fe^{3+}$ triggers $Ca^{2+}$ release from the endoplasmic reticulum by stimulations of InsP$_3$R. In conclusion, we discovered that iron ions trigger astrocytic $Ca^{2+}$ signalling by acting through two distinct molecular cascades.

## Results

**$Fe^{2+}$/$Fe^{3+}$ trigger [$Ca^{2+}$]$_i$ increase in cortical astrocytes in vitro and in vivo.** Exposure of cultured astrocytes to $Fe^{2+}$ leads to a gradual increase in cytoplasmic $Fe^{2+}$ concentration as revealed by quenching of Fura-2 (Supplementary Figure 1). Washout of $Fe^{2+}$ leads to a full recovery of Fura-2 fluorescence; these data indicate the existence of plasmalemmal $Fe^{2+}$ transport system as well as absence of non-specific damaging effect of ionised iron on the cellular membrane. We analysed effects of $Fe^{2+}$ and $Fe^{3+}$ $Ca^{2+}$ dynamics in astrocytes in primary cultures and in vivo in GFAP-eGFP transgenic mice (Fig. 1). In the primary cultured astrocytes, administration of either FeSO$_4$ ($Fe^{2+}$) or ferric ammonium citrate-TF ($Fe^{3+}$) increased [$Ca^{2+}$]$_i$ in concentration-dependent manner, albeit with different kinetics. In the presence of $Fe^{2+}$ an increase in [$Ca^{2+}$]$_i$ demonstrated prominent plateau, whereas $Fe^{3+}$ triggered transient, relatively rapidly decaying [$Ca^{2+}$]$_i$ response (Fig. 1a).

When imaging cortical astrocytes in vivo (the cells were identified by specific eGFP fluorescence) we found that addition of either $Fe^{2+}$ or $Fe^{3+}$ for 30 s induced transient [$Ca^{2+}$]$_i$ increase (Fig. 1b). Administration of $Fe^{2+}$ increased fluorescent intensity of Rhod-2 to 456.30% ± 18.46% ($n = 10$, $p < 0.0001$) whereas $Fe^{3+}$ increased the peak of fluorescent signal to 308.50% ± 13.01% ($n = 10$, $p < 0.0001$) of the basal value. In experiments in vitro we characterised concentration-dependence of [$Ca^{2+}$]$_i$ responses: the EC$_{50}$ was around 0.4–0.6 μM for both $Fe^{2+}$ and $Fe^{3+}$ (Fig. 1c).

**DMT1 and TFR mediate $Fe^{2+}$ and $Fe^{3+}$ uptake.** As mentioned above, $Fe^{2+}$ uptake could be mediated by plasmalemmal transporter DMT1, whereas $Fe^{3+}$ is mainly accumulated in TF-bound form by TFRs (Fig. 2a). Immunostaining of cortical tissue preparations and primary cultured astrocytes demonstrated co-localisation of DMT1 and TFR with astroglial GFAP-positive profiles (Fig. 2b). In the cortical tissue both DMT1 and TFR are preferentially localised at perivascular endfeet. Meanwhile, expression of specific DMT1 and TFR mRNA was also detected in the freshly isolated and sorted astrocytes and neurones, as well as and in the cerebral tissue (Fig. 2c).

To reveal the contribution of DMT1 and TFR to $Fe^{2+}$/ $Fe^{3+}$-induced [$Ca^{2+}$]$_i$ dynamics, we inhibited expression of DMT1 or TFR using siRNA duplex chains. The representative western blots demonstrating the efficacy of knockdown are shown in Fig. 2d. When compared to the control group, the DMT1 siRNA reduced expression of DMT1 to 9.52% ± 2.58% ($n = 6$, $p < 0.0001$), whereas treatment with TFR siRNA decreased TFR levels to 7.84% ± 2.10% ($n = 6$, $p < 0.0001$) from the control values. Administration of $Fe^{2+}$ to DMT1-deficient astrocytes failed to induce any changes in [$Ca^{2+}$]$_i$. At the same time $Fe^{2+}$ induced robust [$Ca^{2+}$]$_i$ elevation in astrocytes treated with siRNA(−) (Fig. 2e). Similarly, $Fe^{3+}$ did not produce [$Ca^{2+}$]$_i$ transients in astrocytes treated with TFR siRNA duplex chains,

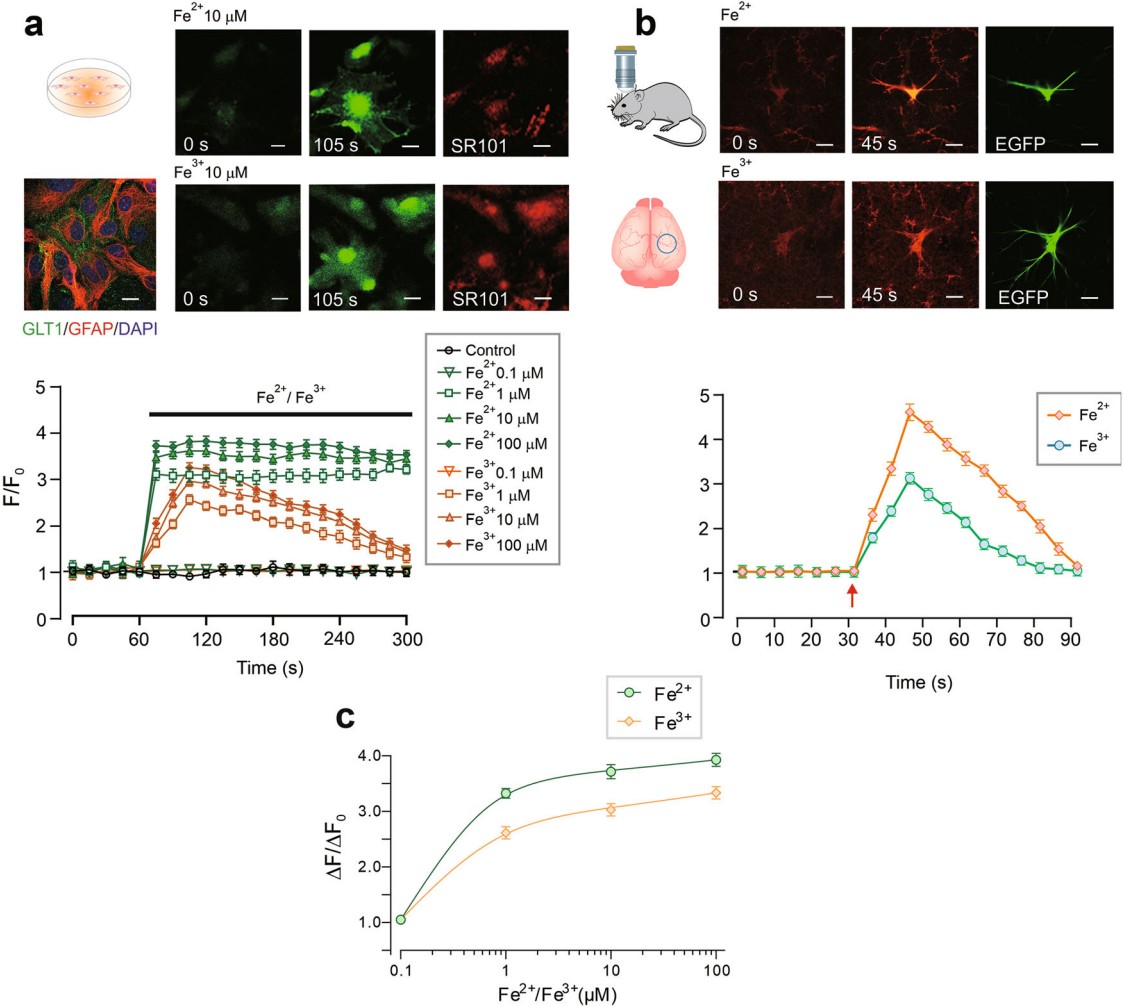

**Fig. 1 Iron ions, $Fe^{2+}$ and $Fe^{3+}$, evoke astrocytic intracellular $Ca^{2+}$ signals in vitro and in vivo. a** Representative images (top panel) and intracellular $Ca^{2+}$ ($[Ca^{2+}]_i$) recordings from primary cultured astrocytes in response to different concentrations of $Fe^{2+}$ or $Fe^{3+}$. Every data point represents mean ± SD, $n = 10$, $p < 0.05$, statistically significant difference from the value of baseline in the same group. The experiment was repeated in 10 different cultures. The representative image on the left shows the primary cultured astrocytes co-stained with GLT1 (green), GFAP (red), and DAPI (blue). Representative Fluo-4 images (green) show astrocytes treated by 10 μM $Fe^{2+}$ or $Fe^{3+}$ at 0 s and 105 s of the recordings, the astrocytic identity was confirmed by co-staining with SR101. Scale bar, 20 μm. **b** Images (top panel) and $[Ca^{2+}]_i$ recordings from cortical astrocytes in GFAP-eGFP transgenic mice using transcranial confocal microscopy. Every data point represents mean ± SD, $n = 10$ (different mice), $p < 0.05$, statistically significant difference from the value of baseline in the same group. Representative Rhod-2 images (red) show astrocytes treated with 100 μM $Fe^{2+}$ or $Fe^{3+}$ at 0 s (baseline) and 45 s (peak of the response); GFAP-eGFP images (green) are shown on the right. Scale bar, 10 μm. **c** Concentration-dependence of the maximal amplitude of $[Ca^{2+}]_i$ responses triggered by $Fe^{2+}$ or $Fe^{3+}$ in cultured astrocytes.

whereas in cells exposed to negative control siRNA $Fe^{3+}$ evoked $[Ca^{2+}]_i$ elevations (Fig. 2e). At the same time $Fe^{3+}$-induced $[Ca^{2+}]_i$ transients were fully preserved in DMT1-deficient astrocytes (Supplementary Figure 2), hence questioning the role of DMT1 in release of $Fe^{3+}$ from endosomes.

In unstimulated astrocytes in culture the DMT1 fluorescence was the highest around the nucleus, suggesting its preferred intracellular localisation. After treatment of cultures with $Fe^{2+}$ for 5 min we observed redistribution of DMT1 from the nuclear region to the plasma membrane (Fig. 2f). The levels of DMT1 were measured in the extracted proteins of the nucleus and cytoplasm (Fig. 2g). After treatment with $Fe^{2+}$ for 5 min, the level of DMT1 in the nuclei decreased to 36.41% ± 4.33% of control group ($n = 10$, $p < 0.0001$), whereas the level of cytoplasmic DMT1 increased to 117.37% ± 5.79% of control group ($n = 10$, $p < 0.0001$) (Fig. 2g).

**Sources of iron-induced $[Ca^{2+}]_i$ mobilisation.** The main sources for $[Ca^{2+}]_i$ increase in astrocytes are (i) $Ca^{2+}$ release from the ER following the opening of InsP$_3$Rs or RyRs, or (ii) plasmalemmal $Ca^{2+}$ influx through either $Ca^{2+}$ permeable channels (such as L-type $Ca^{2+}$ channels or TRP channels) or NCX operating in the reverse mode (Fig. 3a) or (iii) combination of some or all of these pathways. To dissect $Ca^{2+}$ sources we first determined the influence of extracellular $Ca^{2+}$ on iron-evoked $[Ca^{2+}]_i$ transients. Removal of $Ca^{2+}$ from the extracellular milieu completely abolished $Fe^{2+}$-induced $[Ca^{2+}]_i$ elevations but left $Fe^{3+}$-evoked $[Ca^{2+}]_i$ transients largely intact (Fig. 3b). This highlights the role for plasmalemmal $Ca^{2+}$ influx in $Ca^{2+}$ signalling triggered by $Fe^{2+}$ and ER $Ca^{2+}$ release for $Ca^{2+}$ signals triggered by $Fe^{3+}$.

Incubation of astrocytes with inhibitor of L-type voltage-gated $Ca^{2+}$ channel nifedipine (10 μM) did not affect $[Ca^{2+}]_i$ responses to $Fe^{2+}$ or to $Fe^{3+}$ (Fig. 3c). In contrast, inhibition of NCX with

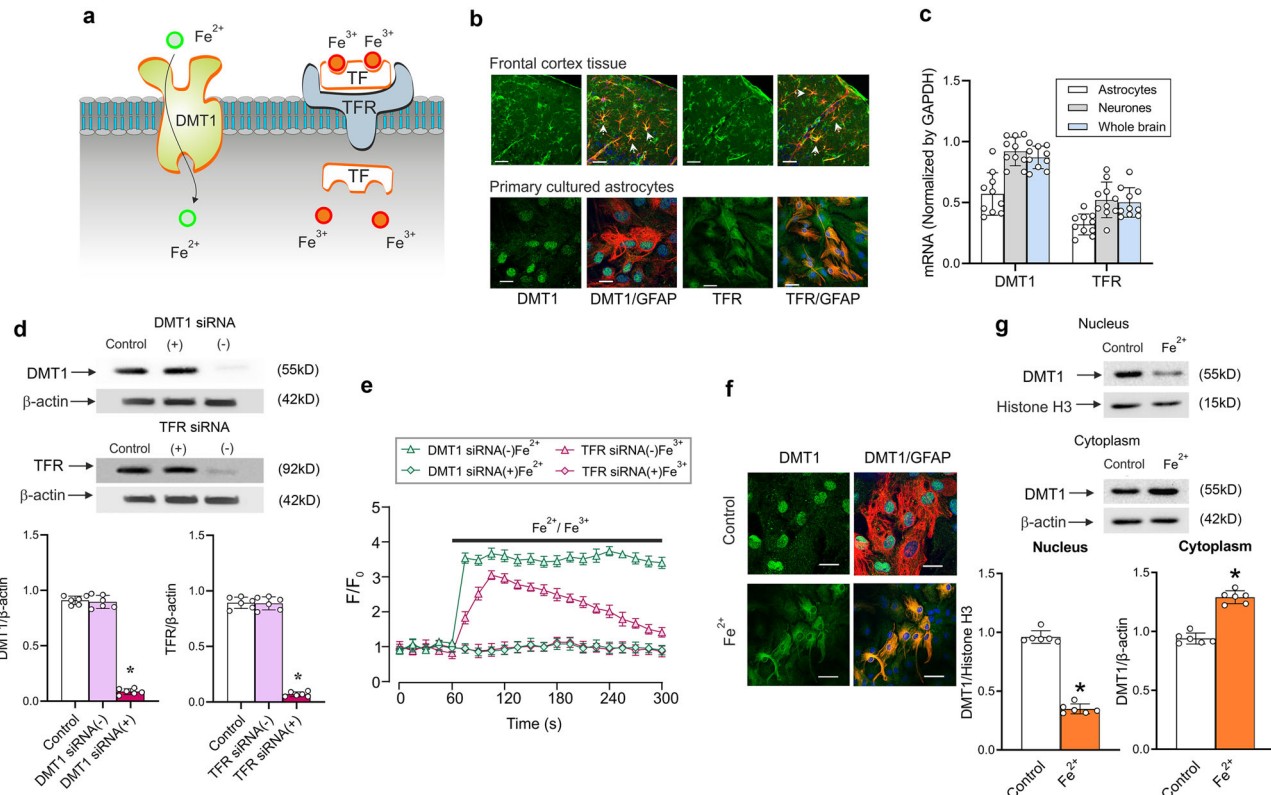

**Fig. 2 Astrocytic expression of DMT1 and TFR. a** Astrocytes accumulate $Fe^{2+}$ through plasmalemmal divalent metal transporter 1 (DMT1/SLC11A2) whereas $Fe^{3+}$ is taken up by internalisation of $Fe^{3+}$-TF-TFR complex. **b** Images of astrocytes in the somato-sensory cortex preparations or in primary culture double-immunolabeled with antibodies against DMT1 or TFR and against GFAP. Scale bar, 20 μm. **c** The mRNA expression of DMT1 and TFR measured by qPCR in astrocytes sorted from GFAP-GFP mice, in neurones sorted from Thy1-YFP mice, and in whole cerebral tissue of wild type mice. **d** Representative western blot bands for DMT1 and TFR in cultured astrocytes treated with sham (Control), siRNA negative control (−) or positive duplex chains (+). The protein levels are shown as the ratio of DMT1 (55 kDa) and β-actin (42 kDa), and TFR (92 kDa) and β-actin. Data represent mean ± SD, $n = 6$. *Indicates statistically significant ($p < 0.05$) difference from the value of baseline in the same group. **e** $[Ca^{2+}]_i$ transients evoked by $Fe^{2+}$ or $Fe^{3+}$ after RNA interference and down-regulation of protein synthesis. After treatment with DMT1 or TFR siRNA negative control (−) or positive duplex chains (+) for 3 days, $[Ca^{2+}]_i$ dynamics in response to $Fe^{2+}/Fe^{3+}$ was monitored. Every data point represents mean ± SD, $n = 10$. **f** Images of astrocytes in primary culture treated with sham (Control) or $Fe^{2+}$ for 5 min and double-immunolabelled with antibodies against DMT1 and against GFAP. Scale bar, 25 μm. **g** Redistribution of DMT1 induced by $Fe^{2+}$ in the extracted proteins of nucleus and cytoplasm. Every data point represents mean ± SD, $n = 10$. *Indicates statistically significant ($p < 0.05$) difference from control group.

10 μM of selective agonist KB-R7943 completely eliminated $[Ca^{2+}]_i$ responses to $Fe^{2+}$, without affecting $Fe^{3+}$-induced $[Ca^{2+}]_i$ transients (Fig. 3d). Thus, two forms of iron, the ferrous and the ferric, mobilise intracellular $Ca^{2+}$ through distinct pathways: $Fe^{2+}$ stimulates $Ca^{2+}$ influx by NCX, whereas $Fe^{3+}$ triggers intracellular $Ca^{2+}$ release. This suggestion was further corroborated by pharmacological inhibition of $InsP_3$ receptors with potent antagonist Xestospongin C (XeC)[38], Exposure of cultured astrocytes to 10 μM of XeC effectively suppressed $[Ca^{2+}]_i$ response to $Fe^{3+}$, without much affecting $Fe^{2+}$-induced $[Ca^{2+}]_i$ transient (Fig. 3e). Finally, treatment with 10 μM ryanodine (which at this concentration inhibits RyRs) somewhat decreased the plateau phase of $Fe^{2+}$-induced $[Ca^{2+}]_i$ transient without modifying $[Ca^{2+}]_i$ response to $Fe^{3+}$ (Fig. 3f).

**DMT1 transports $Fe^{2+}$, which inhibits NKA, increases $[Na^+]_i$ and reverses NCX.** Experiments described above have demonstrated that $Fe^{2+}$, after being transported into the cell by DMT1, leads to a reversal of the NCX, which results in $Ca^{2+}$ influx. The NCX reversal in astrocytes is triggered by an increase in $[Na^+]_i$. Such an increase may originate either from the activation of plasmalemmal $Na^+$ entry or from inhibition of the NKA, which maintains basal $[Na^+]_i$[28,30]. The activity of NKA was suppressed

by exposure to 10 μM $Fe^{2+}$ to 82.40 ± 5.74% ($n = 10$, $p < 0.0001$) of the control. Exposure to 100 nM of the specific NKA inhibitor, ouabain, reduced NKA activity to 72.30 ± 5.91% ($n = 10$, $p < 0.0001$) of the control (Fig. 4a). When 10 μM $Fe^{2+}$ and 100 nM ouabain were added together, the NKA activity was reduced to 71.80 ± 7.81% ($n = 10$, $p < 0.0001$) (Fig. 4a).

Inhibition of NKA in astrocytes results in a substantial elevation in $[Na^+]_i$. When monitoring $[Na^+]_i$ in cultured astrocytes with $Na^+$-sensitive probe SBFI we found that both $Fe^{2+}$ (10 μM) and ouabain (100 nM) triggered rapid and substantial elevation of $[Na^+]_i$ (Fig. 4b). These changes in $[Na^+]_i$ were paralleled by $[Ca^{2+}]_i$ dynamics. Exposure of astrocytes to $Fe^{2+}$, or ouabaine or mixture of $Fe^{2+}$ and ouabain caused $[Ca^{2+}]_i$ elevation (Fig. 4c). When $Fe^{2+}$ was applied in the presence of ouabain it failed to change $[Ca^{2+}]_i$ (Fig. 4d); at the same time application of $Fe^{3+}$ in the presence of ouabain still triggered additional $[Ca^{2+}]_i$ elevation (Fig. 4d).

**$Fe^{2+}$-induced $Ca^{2+}$ mobilisation is associated with caveolae.** Treatment of cultured astrocytes with interfering Cav3 siRNA duplex chains decreased the level of Cav3 to 7.41 ± 4.32% ($n = 6$, $p < 0.0001$) of the control (Fig. 4e). An in vitro knock-down of Cav3 significantly reduced amplitudes of $Fe^{2+}$-induced $[Ca^{2+}]_i$

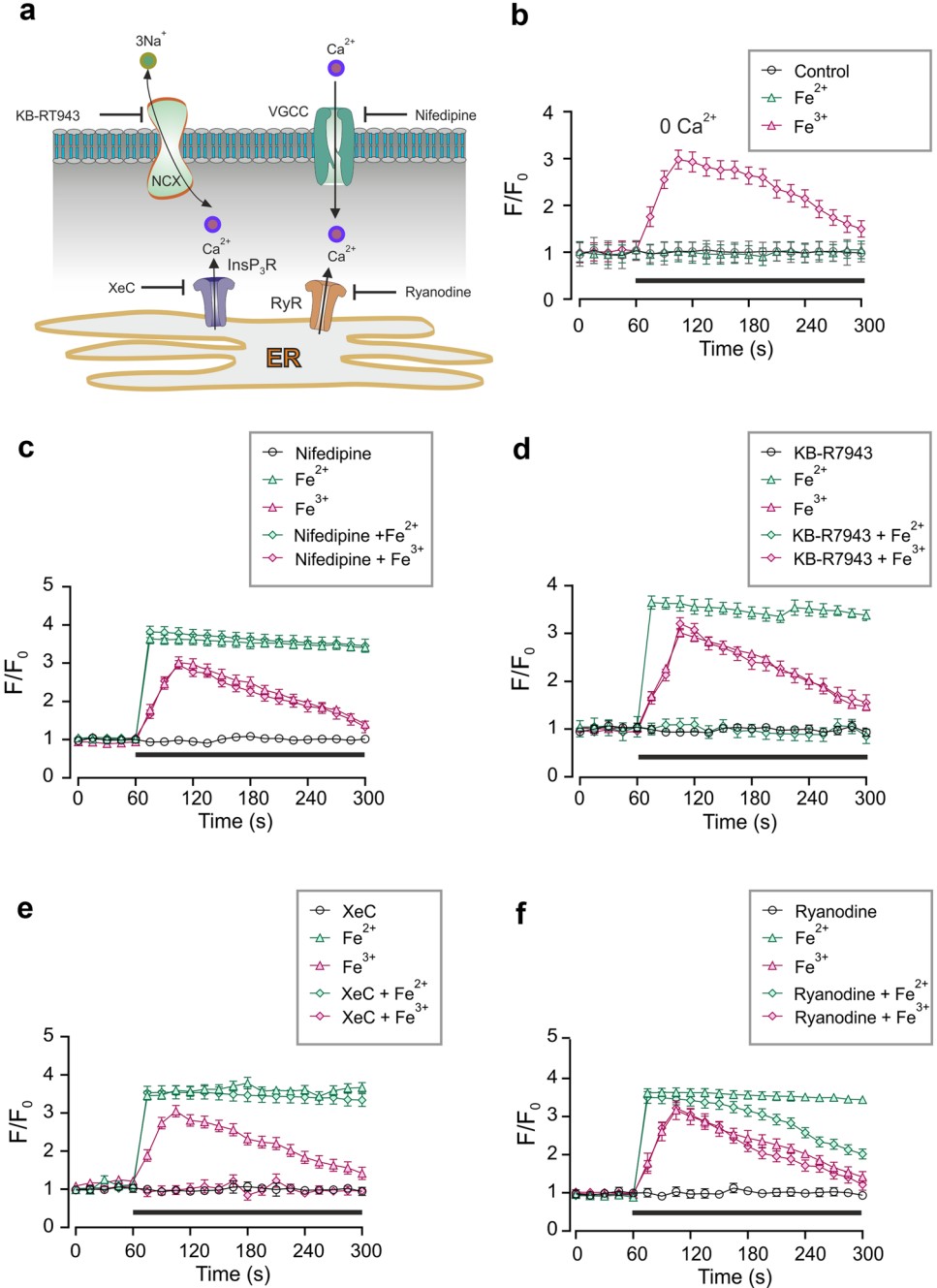

**Fig. 3 $Ca^{2+}$ sources for $Fe^{2+}$ or $Fe^{3+}$-induced $[Ca^{2+}]_i$ transients. a** Possible pathways mediating $Ca^{2+}$ influx into the cytosol. **b** $[Ca^{2+}]_i$ recordings performed in the $Ca^{2+}$-free extracellular medium; removal of external $Ca^{2+}$ abolished $Fe^{2+}$ evoked $[Ca^{2+}]_i$ increase; conversely $[Ca^{2+}]_i$ transient evoked by $Fe^{3+}$ remains intact. **c** Pre-treatment with 10 μM nifedipine affects neither $Fe^{2+}$ nor $Fe^{3+}$-induced $[Ca^{2+}]_i$ responses. **d** Pre-treatment with NCX inhibitor KB-R7943 at 10 μM suppresses $[Ca^{2+}]_i$ response to $Fe^{2+}$ but not to $Fe^{3+}$. **e** Pre-treatment with $InsP_3$ receptor inhibitor XeC at 10 μM suppresses $[Ca^{2+}]_i$ response to $Fe^{3+}$ but not to $Fe^{2+}$. **f** Pre-treatment with 10 μM ryanodine (Ry), the inhibitor of ryanodine receptors, affects only plateau phase of $Fe^{2+}$-induced $[Ca^{2+}]_i$ response. In b-f, every data point represents mean ± SD, $n = 10$.

increase; maximal increase in Fluo-4 $F/F_0$ after Cav3 knockdown was $242.00 \pm 16.44\%$ ($n = 10$, $p < 0.0001$). In control astrocytes treated with siRNA($-$) the amplitude of $Fe^{2+}$-induced $[Ca^{2+}]_i$ increase reached $363.34 \pm 11.62\%$ ($n = 10$, $p < 0.0001$, Fig. 4f). To further analyse the effects of Cav3 on $Fe^{2+}$-induced $[Ca^{2+}]_i$ dynamics the levels of relevant proteins were measured in the extracted caveolae (Fig. 5a). As shown in Fig. 5B, exposure to 10 μM $Fe^{2+}$ for 5 min significantly increased the level of DMT1 to $293.24 \pm 24.89\%$ ($n = 10$, $p < 0.0001$) of the control values.

After pre-treatment with Cav3 siRNA duplex chains, $Fe^{2+}$ increased the level of DMT1 only to $195.77 \pm 20.19\%$ ($n = 10$, $p < 0.0001$) of control group (Fig. 5b). The levels of NCX1 and NKA were similarly affected by $Fe^{2+}$ and the knocking down of Cav3. Exposure to $Fe^{2+}$ increased the level of NCX1 and NKA to $248.71 \pm 19.58\%$ ($n = 10$, $p < 0.0001$) and $263.66 \pm 25.93\%$ ($n = 10$, $p < 0.0001$) of the controls, after knocking down Cav3, $Fe^{2+}$ elevated the level of NCX1 and NKA only to $172.96 \pm 11.76\%$ ($n = 10$, $p < 0.0001$) and $200.86 \pm 18.14\%$ ($n = 10$, $p < 0.0001$) of

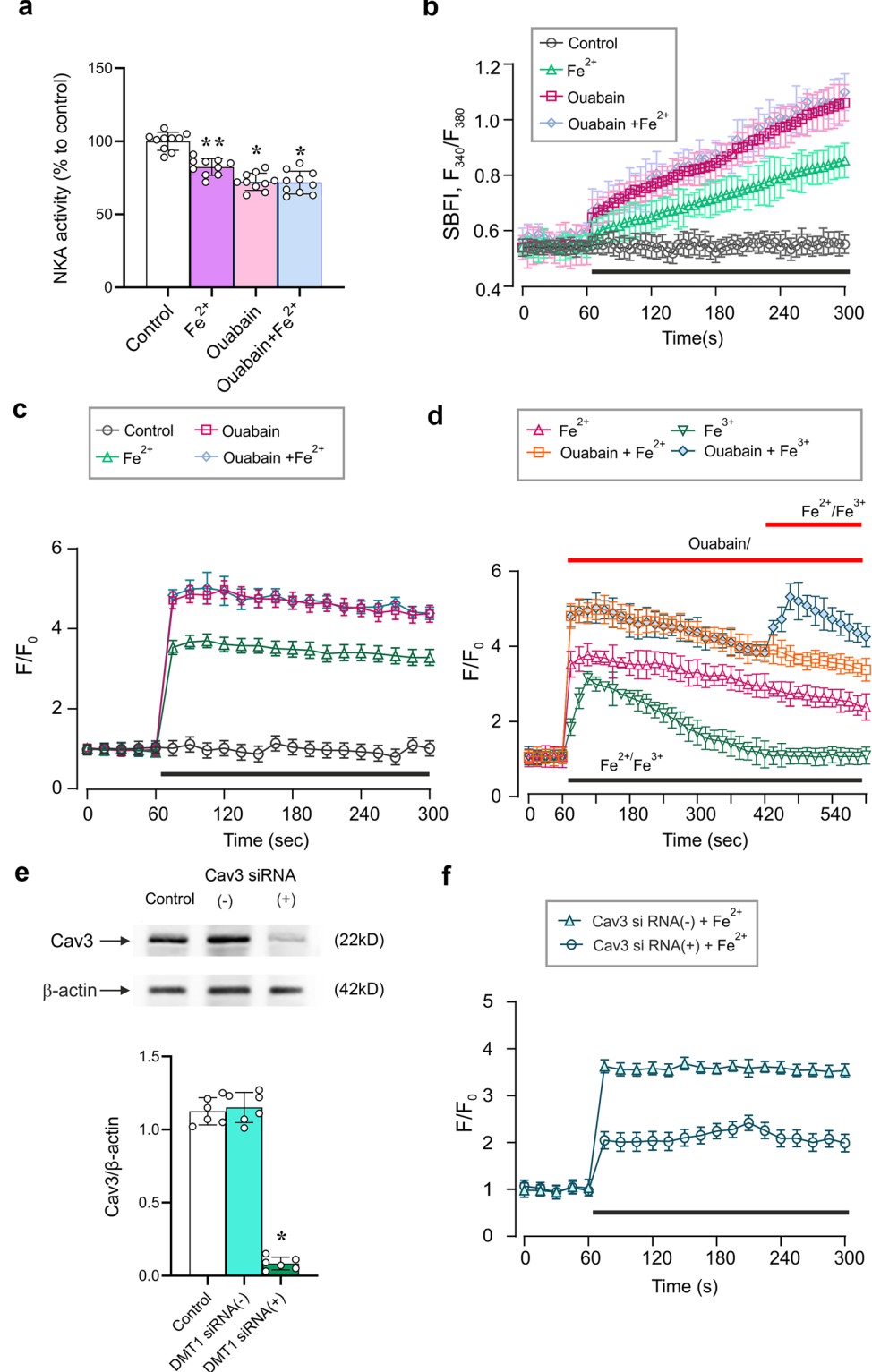

**Fig. 4 Fe$^{2+}$ reverses NCXs by inhibiting NKA. a** NKA activity in the presence of Fe$^{2+}$ and ouabain measured by ELISA; OD value was normalised to controls. Data represent mean ± SD, $n = 10$. *Indicates statistically significant ($p < 0.05$) difference from control group; **indicates statistically significant (p < 0.05) difference from any other group. **b** [Na$^+$]$_i$ recordings from cultured astrocytes challenged with Fe$^{2+}$ and ouabain as indicated on the graph (**c**) and (**d**) [Ca$^{2+}$]$_i$ recordings from cultured astrocytes challenged with Fe$^{2+}$, Fe$^{3+}$ and ouabain as indicated on the graph. For b, c and d every data point represents mean ± SD, $n = 10$. **e** Representative western blot bands for Cav3 in cultured astrocytes treated with sham (Control), siRNA negative control (−), or positive duplex chains (+) to down-regulate Cav3 expression. Protein values are shown as the ratio of Cav3 (22 kDa) to β-actin (42 kDa). Data represent mean ± SD, $n = 6$. *Indicates statistically significant ($p < 0.05$) difference from any other group. **f** Cav3 RNA interference suppresses Fe$^{2+}$-induced [Ca$^{2+}$]$_i$ response. Every data point represents mean ± SD, $n = 10$.

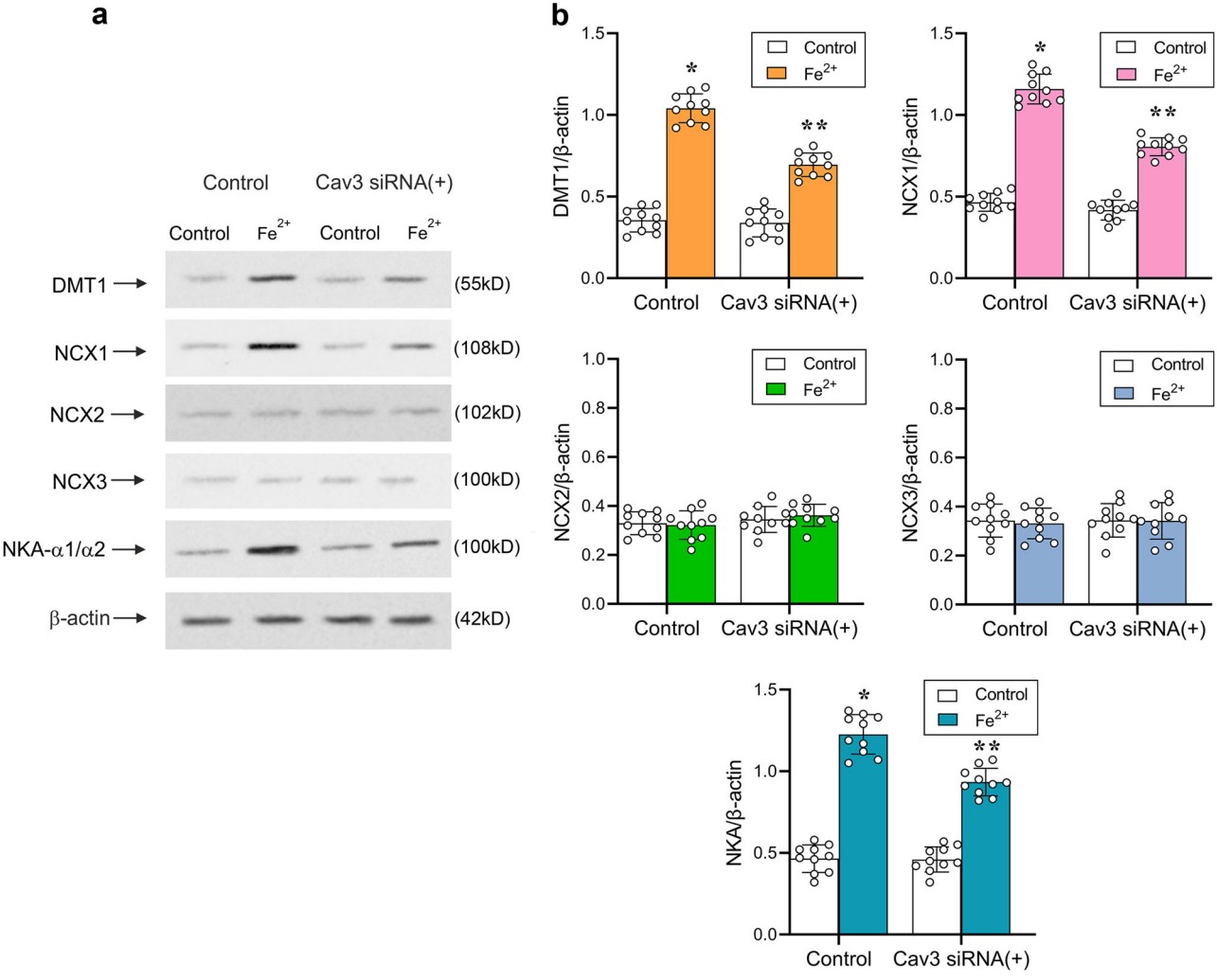

**Fig. 5 Caveolae integrate DMT1, NKA and NCX. a** Representative protein western blot bands for DMT1, NCX1-3 and NKA-α1/α2 in caveolae membranes extracted from cultured astrocytes. Astrocytes were treated with sham (Control), with siRNA negative control (−) or with Cav3 duplex chains (siRNA(+)) for 3 days. **b** Protein levels are shown as the ratio of DMT1 (55 kDa) and β-actin (42 kDa), NCX1 (108 kDa) and β-actin, NCX2 (102 kDa) and β-actin, NCX3 (100 kDa) and β-actin, NKA-α1/α2 (100 kDa) and β-actin (42 kDa). Data represent mean ± SD, $n = 6$. *Indicates statistically significant ($p < 0.05$) difference from any other group; **indicates statistically significant ($p < 0.05$) difference from control plus $Fe^{2+}$ group or Cav3 siRNA(+) plus Ctrl group.

control values (Fig. 5b). Of note, $Fe^{2+}$ did not affect the levels of NCX2 and NCX3 (Fig. 5b).

**$Fe^{3+}$ triggers $Ca^{2+}$ release through stimulation of PLC and increase in InsP$_3$ production.** As shown in Fig. 3e, inhibition of InsP$_3$ receptors with XeC suppressed $Fe^{3+}$-induced $[Ca^{2+}]_i$ mobilisation. We therefore analysed effects of $Fe^{3+}$ on the InsP$_3$ signalling cascade in cultured astrocytes. Incubation of astrocytes with $Fe^{3+}$ increased the level of InsP$_3$ in cell lysates to 74.10 ± 8.14 ng/ml ($n = 10$, $p < 0.0001$), from the control InsP$_3$ level of 24.40 ± 6.35 ng/ml. In cells treated with TF alone the InsP$_3$ level was 28.00 ± 12.62 ng/ml ($n = 10$, $p = 0.4309$) (Fig. 6a). Subsequently we analysed the links between scaffolding/signalling protein Dab2 and $Fe^{3+}$-induced $Ca^{2+}$ signalling. We suppressed expression of two isoforms of Dab2 by siRNA duplex chains (Fig. 6b). After RNA interfering, expression of 96 kD and 67 kD Dab2 isoforms decreased, respectively, to 5.81 ± 3.56% ($n = 6$, $p < 0.0001$) and to 11.88 ± 6.51% ($n = 6$, $p < 0.0001$) of the control values (Fig. 6b). The knock-down of Dab2 rendered $Fe^{3+}$ ineffective: exposure of Dab2-deficient astrocytes to $Fe^{3+}$ did not affect InsP$_3$ production (control: 31.60 ± 12.51 ng/ml ($n = 10$);

Dab2 knockdown: 30.60 ± 10.42 ng/ml ($n = 10$, $p = 0.1253$; Fig. 6c).

When Dab2-deficient astrocytes were challenged with $Fe^{3+}$, no $[Ca^{2+}]_i$ increase was recorded (Fig. 6d). Similarly, after inhibition of the PLC with U-73122, application of $Fe^{3+}$ did not change $[Ca^{2+}]_i$ (Fig. 6d). Hence, we may surmise that uptake of $Fe^{3+}$ through TFR requires Dab2 protein; after entering the cytosol $Fe^{3+}$ activates the PLC, which produces InsP$_3$ that triggers InsP$_3$-induced $Ca^{2+}$ release from the ER (Fig. 6e).

## Discussion
In this paper, we describe previously unknown effects of iron ions on cellular $[Ca^{2+}]_i$ in astrocytes. Administration of either $Fe^{2+}$ or $Fe^{3+}$ triggered a concentration-dependent increase in $[Ca^{2+}]_i$ with EC$_{50}$ of about 0.4–0.6 μM. We performed an in depth analysis of the mechanisms underlying iron transport and iron induced $Ca^{2+}$ signalling. We also demonstrated that, contrary to the previous beliefs, astrocytes express functional TFR in vitro and in vivo thus allowing accumulation of $Fe^{3+}$.

In the brain, majority of cells express a full complement of proteins responsible for iron homoeostasis, including TFR and DMT1 for iron uptake, heavy and light chains ferritin for iron

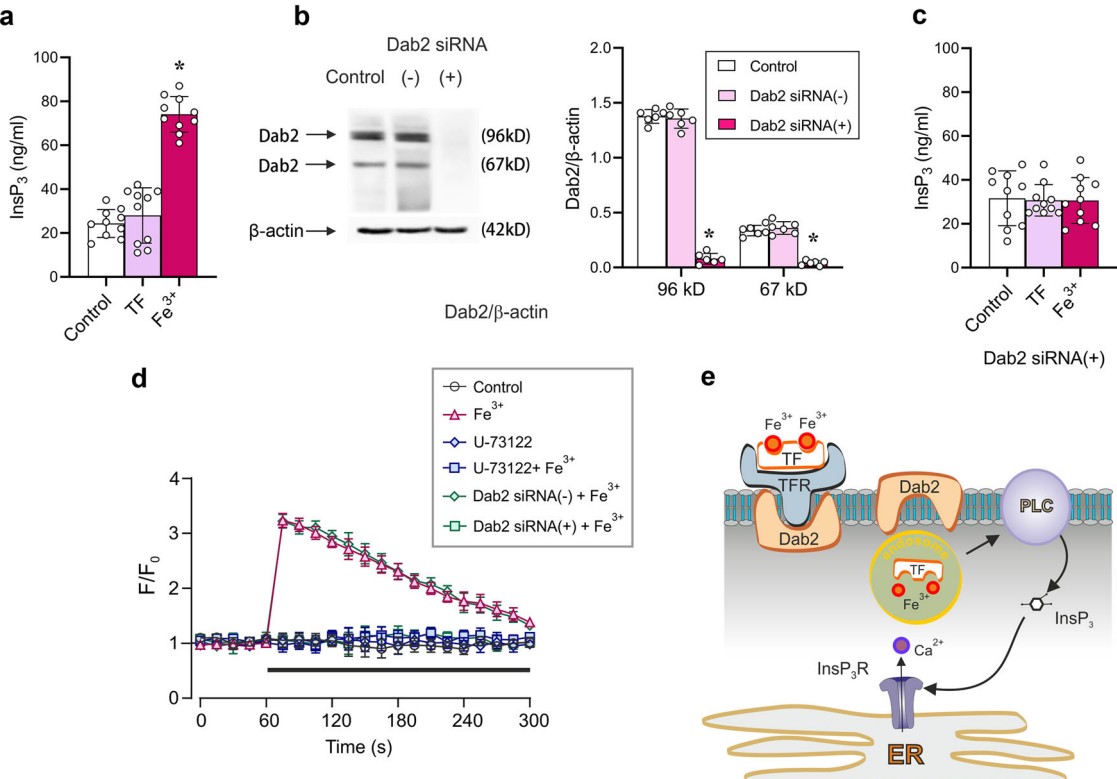

**Fig. 6 Mechanisms of $Fe^{3+}$ induced $Ca^{2+}$ signalling. a** Incubation with $Fe^{3+}$ increases $InsP_3$ level in cultured astrocytes; $InsP_3$ was measured with ELISA; data are presented as mean ± SD, $n = 10$. *Indicates statistically significant ($p < 0.05$) difference from any other group. **b** Representative protein bands for two isoforms (96 KDa and 67 KDa) of Dab2 in cultured astrocytes treated with sham (Control), siRNA negative control (−) or positive duplex chains (+). Protein values are shown as the ratio of 96 kDa isoform and β-actin (42 kDa), and the ratio of 67 kDa isoform and β-actin. Data represent mean ± SD, $n = 6$. *Indicates statistically significant ($p < 0.05$) difference from any other group. **c** $Fe^{3+}$-dependent changes in the $InsP_3$ after treatment with Dab2 siRNA duplex chains. After treatment with Dab2 siRNA duplex chains (+) for 3 days, the primary cultured astrocytes were treated with serum-free medium (Control), TF (as negative control) or 10 μM $Fe^{3+}$-TF ($Fe^{3+}$) for 5 minutes, the level of $InsP_3$ was measured by ELISA and shown as mean ± SD, $n = 10$. **d** $Fe^{3+}$ -induced $[Ca^{2+}]_i$ responses are $Dab^2$ and PLC dependent. Treatment with Dab2 siRNA duplex chains (+) as well as with 10 μM U-73122 (PLC inhibitor) suppressed $Fe^{3+}$-induced $[Ca^{2+}]_i$ responses. Every data point represents mean ± SD, $n = 10$. **e** Mechanisms of $Fe^{3+}$-induced $Ca^{2+}$ signalling in astrocytes. The uptake of $Fe^{3+}$ is mediated TFR, TFR internalization requires Dab2; when in the cytosol $Fe^{3+}$ activated PLC thus stimulating InsP3-induced $Ca^{2+}$ release from the ER.

sequestration, cytosolic iron exporter ferroportin 1 (FPN1), and iron regulatory protein 1 and 2 (IRP1and IRP2) for regulating intracellular iron homoeostasis[39,40]. Glial cells, and astrocytes in particular, store up to 75% of ionised iron in the CNS[41], and protect the brain against iron overloads[42]. Transmembrane transport of iron in astrocytes has been identified, but was not studied in details. There is a general agreement on the primary role of plasmalemmal divalent metal transporter 1, DMT1/SLC11A2, which selectively transports $Fe^{2+}$; the DMT1 was detected in astrocytes in culture and there is evidence indicating its presence in astroglial endfeet in situ[43–45]. In addition, $Fe^{2+}$ was suggested to enter reactive astrocytes by diffusion through transient receptor potential canonical (TRPC) channels[42]. Expression of TF-$Fe^{3+}$-transporting TFR has been noted in astrocytes in culture[21,46]; it is, nonetheless, generally believed that astrocytes in vivo are not in a possession of TFR and hence cannot accumulate $Fe^{3+}$ [47–49]. This conclusion, however, has been made on the basis of rather limited investigations[14,46]; while expression of TFR-specific mRNA was detected in astrocyte transcriptome[50]. In our study we confirmed expression of DMT1, at mRNA and protein levels as well as by immunostaining, in acutely isolated astrocytes, in astroglial primary culture and in situ in cortical tissue; the DMT1 was particularly enriched in the endfeet (Fig. 2b–d). We also detected astroglial expression of TFR at mRNA level in the transcriptome of acutely isolated and

FACS-sorted astrocytes (Fig. 2c). We further confirmed expression of TFR in astrocytes at a protein level and in immunohistochemical analysis of astrocytes in culture and in cortical preparations (Fig. 2b–d). In the cortical tissue TFR labelling was concentrated in perivascular astrocytic endfeet (Fig. 2b–d).

Not much is known about the links between ionised iron and $Ca^{2+}$ signalling in the cellular elements of the CNS. In the literature, we found only a single example of $Fe^{3+}$-induced $[Ca^{2+}]_i$ transient in cultured hippocampal neurones[51]. To the best of our knowledge here we present the first recordings of $Fe^{2+}/Fe^{3+}$ induced $Ca^{2+}$ signals in astrocytes. Both ions evoked $[Ca^{2+}]_i$ elevation in primary cultured astrocytes and in EGFP-labelled astrocytes in the cortices of alive animals studied with transcranial confocal microscopy. Both ions acted in the low μM range of concentrations, however the kinetics of respective $[Ca^{2+}]_i$ transients are different. Exposure of cultured astrocytes to $Fe^{2+}$ triggered rapid $[Ca^{2+}]_i$ increase with long-lasting plateau; the $[Ca^{2+}]_i$ barely declined in the presence of $Fe^{2+}$. In contrast, $Fe^{3+}$-induced transient elevation of $[Ca^{2+}]_i$ recovers to the baseline within ~ 200–300 s in the presence of $Fe^{3+}$ (Figs. 1b and 2e). These distinct kinetics reflect different signalling cascades activated by iron ions.

The $Fe^{2+}$-induced $[Ca^{2+}]_i$ responses require DMT1: in vitro knockdown of DMT1 expression with silencing mRNA completely eliminates $Ca^{2+}$ signal (Fig. 2). The $Fe^{2+}$-induced $[Ca^{2+}]_i$

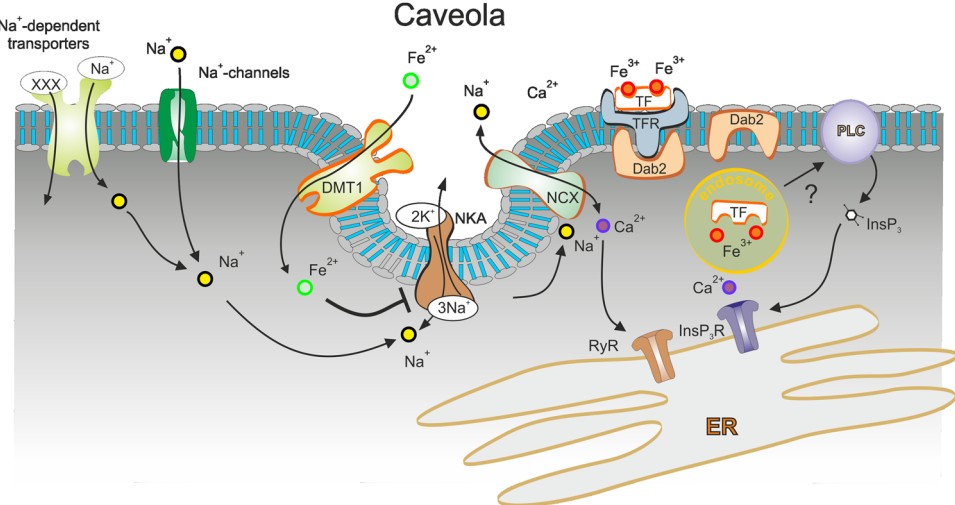

**Fig. 7 Mechanisms of $Fe^{2+}$ and $Fe^{3+}$ -induced intracellular $Ca^{2+}$ signalling.** Uptake of $Fe^{2+}$ is mediated by DMT1 and uptake of $Fe^{3+}$ is mediated by TFR. After entering the cytosol, $Fe^{2+}$ inhibits the activity of NKA, which causes an increase in $[Na^+]_i$ due to an unopposed influx of $Na^+$ through $Na^+$-dependent transporters or $Na^+$-channels. Increase in $[Na^+]_i$ switches the NCX into the reverse mode, which results in $Ca^{2+}$ influx. This influx also activates $Ca^{2+}$-induced $Ca^{2+}$ release through RyR that contributes to the plateau phase of $[Ca^{2+}]_i$ response. Accumulation of $Fe^{2+}$ also promotes the formation of the functional unit of DMT1, NCX1 and NKA in the caveolae by recruiting Cav3. Uptake of $Fe^{3+}$ proceeds through Dab2-assisted internalisation; after entering the cytosol, $Fe^{3+}$ activates the PLC and stimulates $InsP_3$-dependent $Ca^{2+}$ release from the ER.

changes originate from plasmalemmal $Ca^{2+}$ entry, because removal of $Ca^{2+}$ from the extracellular medium inhibits $[Ca^{2+}]_i$ response. Finally, $Fe^{2+}$-induced $[Ca^{2+}]_i$ signals require NCX, as pharmacological inhibition of the latter effectively suppresses $[Ca^{2+}]_i$ elevation (Fig. 3). These data indicate that $Fe^{2+}$, after being accumulated in the astrocyte, switches the NCX into the reverse mode of operation thus generating $Ca^{2+}$ influx into the cell in exchange for $Na^+$. This scenario requires increase in astroglial $[Na^+]_i$, which readily reverses the NCX[28,52]. An increase in $[Na^+]_i$ is likely to result from the inhibition of NKA, which represents the major $Na^+$ efflux mechanism in astrocytes[30]. Activity of NKA was effectively suppressed by $Fe^{2+}$, and probing astrocytes with $Na^+$-sensitive indicator SBFI revealed $Fe^{2+}$-induced $[Na^+]_i$ elevation (Fig. 4). In the presence of NKA inhibitor ouabain $Fe^{2+}$-induced $[Ca^{2+}]_i$ responses were eliminated thus corroborating the central role of NKA and NCX in $Fe^{2+}$-induced $Ca^{2+}$ signalling (Figs. 4, 7). Of note, exposure to $Fe^{2+}$ initiated rapid redistribution of DMT1 from nucleus into cytoplasm (Fig. 2g) and arguably to the plasmalemma thus increasing astrocyte capacity for iron uptake.

The mechanism of $Fe^{3+}$-induced $Ca^{2+}$ signalling is associated with intracellular $Ca^{2+}$ release. The $Fe^{3+}$-induced $[Ca^{2+}]_i$ responses are preserved in $Ca^{2+}$-free extracellular solution while being blocked by XeC (inhibitor of $InsP_3$ receptor) and by U-73122 (inhibitor of PLC) thus revealing the central role for $InsP_3$-mediated ER $Ca^{2+}$ release. Initiation of this signalling cascade requires transmembrane transport of $Fe^{3+}$: in vitro knockdown of TFR eliminates $Fe^{3+}$-evoked $[Ca^{2+}]_i$ dynamics. The internalisation of $TF-Fe^{3+}-TFR$ complex also requires functional Dab2 protein. This protein is a multi-modular scaffold protein with signalling roles in ion homoeostasis, inflammation and receptors internalization[53]. Ablation of Dab2 in astrocytes with specific siRNA interrupts signalling chain and blocks $Fe^{3+}$-dependent $Ca^{2+}$ signalling (Figs. 6 and 7).

Import of $TF-Fe^{3+}-TFR$ complex involves endocytosis; acidic (pH ~5.6) environment of the endosome achieved through the action of ATP-dependent $H^+$ pumps[54], reduces the affinity of TF to $Fe^{3+}$ and the latter is released[55]. Subsequently, an endosomal reductase reduces $Fe^{3+}$ to $Fe^{2+}$, and $Fe^{2+}$ can be transported into the cytosol by DMT1 or by ZIP8 or ZIP14 Zn transporters[56]. Our data question the DMT1 pathway because siRNA knockout of DMT1 did not affect $Fe^{3+}$-induced $Ca^{2+}$ signalling; the detailed mechanism of $Fe^{3+}$ transport in astrocytes remains to be fully characterised.

Caveolae are specific plasmalemmal structures that form functional microdomains involved in various signalling, endocytotic and transporting events[57]. Caveolae and their main structural and regulatory proteins Caveolin-1,2,3 are present in astrocytes (with predominant expression of Cav3); astrocytic caveolae contribute to signal transduction, formation of signalling protein complexes and are involved in action of various neuroactive substances and drugs[58,59]. Caveolae are known to form functional $Ca^{2+}$ signalling units, establish links between $Ca^{2+}$ channels and various transports and may contribute to formation of plasmalemmal/ER functional domains operational in astrocytes[60]. We found that down-regulation of expression of Cav3 in cultured astrocytes substantially reduced the amplitude of $Fe^{2+}$-evoked $[Ca^{2+}]_i$ responses. We suggest therefore that Cav3 and caveolae integrate DMT1, NKA and NCX into a single $Ca^{2+}$ signalling unit (Fig. 7); moreover exposure to iron potentiates formation of such units.

Iron homoeostasis is of fundamental importance for cells, tissues and organisms, as iron contributes to a wide range of vital biological pathways[61]. The brain contains high concentrations of bound and free iron, which participates in multiple processes from energy production to synaptic transmission[41]. Iron overload and failures in iron homoeostatic triggers neurotoxicity and is implicated in brain diseases[48]. Genetic mutations of iron regulatory proteins (the key elements of iron homoeostasis) result in iron deposition in the brain with subsequent neurodegeneration characteristic for aceruloplasminemia[62] and neuroferritino-pathy[63]. In aceruloplasminemia patients, iron overload has been observed in astrocytes, particularly in the basal ganglia[64]. Ceruloplasmin oxidises $Fe^{2+}$ to $Fe^{3+}$ in order to generate the oxidised form of iron that can bind to extracellular transferrin; in astrocytes ceruloplasmin is also required for iron export[64]. Thus, in aceruloplasminemia, iron entering the CNS as ferrous might escape oxidation; hence cells exposed to the excess of $Fe^{2+}$ may readily become iron overloaded[65]. Similarly, abnormal iron accumulation has been characterised in neurodegenerative diseases including

Alzheimer's disease, Parkinson disease, amyotrophic lateral sclerosis and Huntington disease to name but a few[48]. Specific class of neurodegeneration with brain iron accumulation (NBIA) has been also categorised in recent years[66].

Based on our data we propose that astrocytes mount the defence against acute iron overload. This defence includes iron accumulation through both DMT1 and TFR, redistribution of DMT1 from intracellular locations to plasmalemma and generation of $Ca^{2+}$ signals, which control astrocytic reactivity. Iron-induced $Ca^{2+}$ signalling is activated at low pathological iron concentrations (>1 μM; while physiological iron concentration in the CSF ranges between 0.3 and 0.75 μM). Importantly, two distinct signalling cascades (DMT1 $Fe^{2+}$ transport, inhibition of NKA and reversal of NCX versus $Fe^{3+}$-TF-TFR transport, activation of PLC and generation of $InsP_3$-induced $Ca^{2+}$ release) distinguish between ferric and ferrous. These distinct pathways may define very different outputs: it is known for example that activation of astroglial $InsP_3$ receptors type II is linked to initiation of reactive astrogliosis[67,68]. In our recent report, exposure to iron was found to stimulate the over-expression of DMT1 in astrocytes and microglia, but not in neurones, which may result in the neuroprotection by glial uptake of excessive iron[13]. Astrogliosis plays important, if not defining role in evolution of many neurological diseases[69]. Our previous experiments have shown that formation of brain deposits of iron up-regulates astroglial expression of TFR and instigates reactive astrogliosis[70]. What characterises the iron-induced reactive phenotype and what is the role of astroglial reactivity in managing excessive iron in the brain remains to be found. In conclusion, our study presented a phenomenon that iron ions ($Fe^{2+}$ and $Fe^{3+}$) directly induce intracellular $Ca^{2+}$ signalling and stimulate astroglial protective mechanisms against iron overload in broad pathological contexts.

## Methods

**Materials**. The culture medium including DMEM and fetal bovine serum were purchased from Gibco Life Technology Invitrogen (Grand Island, NY, USA). Oligo-fectamine, MEMI, fluo-4 AM, sodium-binding benzofuran isophthalate (SBFI) AM, G-agarose bead, TFR antibody, β-actin antibody, GFAP antibody, DMT1 antibody and DMT1 siRNA duplex chains were from Thermo Fisher Scientific (Waltham, MA USA); siRNA duplex chains of TFR, NCX1-3, Cav-3 and Dab2, NCX2 antibody, $Na^+/K^+$-ATPase alpha1/2 antibody and the secondary antibodies were bought from Santa Cruz Biotechnology (Santa Cruz, CA, USA). NCX1 antibody, NCX3 antibody and native mouse apo-transferrin (apo-TF; i.e., iron-free) were from Abcam (Cambridge, MA, USA). Donkey serum, xestospongin C (Xe-C), nifedipine, ferrous sulfate heptahydrate ($FeSO_4$), sulforhodamine 101 (SR101) and ferric ammonium citrate were purchase from Sigma-Aldrich (St. Louis, MO, USA). Ryanodine and KB-R7943 were purchased from Calbiochem (La Jolla, CA, USA). Secondary antibody staining with donkey anti-mouse or anti-rabbit Cy-2/3 were from Jackson Immuno-Research (West Grove, PA, USA). Primary antibody of histone H3 was purchased from EarthOx (Millbrae, CA, USA).

**Animals**. The C57BL/6 mice, FVB/N-Tg(GFAP-eGFP)14Mes/J and B6.Cg-Tg (Thy1-YFP)HJrs/J transgenic mice were all purchased from the Jackson Laboratory (Bar Harbor, ME, USA). The animals were raised in standard housing conditions (22 ± 1 °C; light/dark cycle of 12/12 h), with water and food available *ad libitum*. All experiments were performed in accordance with the US National Institutes of Health Guide for the Care and Use of Laboratory Animals (NIH Publication No. 8023) and its 1978 revision, and all experimental protocols were approved by the Institutional Animal Care and Use Committee of China Medical University, No. [2019]059.

**Primary culture of astrocytes**. Astrocytes were cultured from newborn mice[71,72]. In brief, the cerebral hemispheres were isolated, dissociated and filtered. Isolated astrocytes were grown in Dulbecco's Minimum Essential Medium (DMEM) with 7.5 mM glucose supplemented with 10% foetal bovine serum. Astrocytes were incubated at 37 °C in a humidified atmosphere of $CO_2$/air (5:95%). The cultures are highly enriched in astrocytes, the purity is >95% as judged by GFAP staining[73].

**Iron treatment**. For preparing $Fe^{3+}$-TF solution, ferric ammonium citrate and mouse apo-TF were incubated at a 2:1 ratio in serum-free culture medium for 1 h at 37 °C[74,75]. The same concentration of apo-TF in the same volume of culture

medium but without $Fe^{3+}$ was used for control treatments. For the $Fe^{2+}$ solution, $FeSO_4$ was freshly dissociated in serum-free culture medium at 37 °C and used immediately, the same volume of serum-free culture medium was used as the control for $Fe^{2+}$ group.

**RNA interfering**. The cultured astrocytes were incubated in DMEM without serum for 12 hours before transfection[73,76,77]. A transfection solution containing 2 μl oligo-fectamine (Promega, Madison, WI, USA), 40 μl MEMI, and 2.5 μl siRNA (DMT1, TFR, NCX1-3, Cav-3 or Dab2) was added to the culture medium in every well for 8 h. In the siRNA-negative control cultures, transfection solution without siRNA was added. Thereafter, DMEM with three times serum was added to the cultures. These siRNA duplex chains were purchased from Santa Cruz Biotechnology (CA, USA).

**Preparation of membrane caveolae**. Cell homogenization and the caveolae preparation from astrocytes was made[78,79]. In brief, primary cultured astrocytes were collected and homogenised in SET (0.315 M sucrose, 20 mM Tris-HCl, and 1 mM EDTA, pH 7.4), and centrifuged for 1 h at 1,000 × g. The pellets were re-solubilised in SET and layered on Percoll (30% in SET) followed centrifugation at 1,000 × g. The pellets were re-homogenised and re-layered to three sucrose density gradient solution (80%, 30% and 5%) with ultra-centrifugation at 175,000 × g. Finally, the purified caveolae were collected and re-suspended in SET.

**Co-immunoprecipitation**. We used technologies of co-immunoprecipitation and subsequent western blotting to check the conjunction level between NCXs and DMT1[76]. After homogenization, protein content was determined by the Bradford method[80] using bovine serum albumin as the standard. For immunoprecipitation of NCX1-3, whole cell lysates (500 μg) were incubated with 20 μg of anti-NCX1, anti-NCX2 or anti-NCX3 antibody for overnight at 4 °C. Then, 200 μl of washed protein G-agarose bead slurry was added, and the mixture was incubated for another 2 hours at 4 °C. The agarose beads were washed three times with cold phosphate buffer solution (PBS) and collected by pulsed centrifugation (5 s in a microcentrifuge at 14,000 × g), the supernatant was drained off, and the beads were boiled for 5 min. Thereafter, the supernatant was collected by pulsed centrifugation, and the entire immunoprecipitates were subjected to 10% sodium dodecyl sulfate (SDS)-polyacrylamide gel electrophoresis (PAGE).

**Cytoplasm and nucleus protein extraction**. The subcellular fraction was analyzed using the protein extraction Kit (P0028, Beyotime, Shanghai, China)[81], according to the manufacturer's protocol. For further western blotting assays, cytoplasmic and nucleic protein extractions were kept under −20 °C.

**Western blotting**. For quantifying expressions of DMT1, TFR, NCX1-3, Cav-3 or Dab2, the samples containing 100 μg of protein were added to slab gels. After transferring to PVDF membranes, the samples were blocked by 10% skimmed milk powder for 1 h, and membranes were incubated overnight with the primary antibodies, specific to either DMT1 at a 1:300 dilution, TFR at a 1:200 dilution, NCX1 at a 1:100 dilution, NCX2 at a 1:200 dilution, NCX3 at a 1:150 dilution, Cav-3 at a 1:200 dilution, Dab2 at a 1:100 dilution, Histone H3 at 1:500 dilution or β-actin at a 1:1000 dilution. After washing, specific binding was detected by horse-radish peroxidase-conjugated secondary antibodies. Images were analysed with an Electrophoresis Gel Imaging Analysis System (MF-ChemiBIS 3.2, DNR Bio-Imaging Systems, Israel). Band density was measured with Window AlphaEase™ FC 32-bit software[72,82].

**Monitoring of $[Ca^{2+}]_i$**. For $[Ca^{2+}]_i$ monitoring and imaging in cultured astrocytes[83,84], after the pre-treatment with or without inhibitors or siRNA duplex chains, the primarily cultured astrocytes were loaded with 5 μM fluo-4-AM, (Thermo Fisher Scientific (Waltham, MA USA)), for 30 min. Fluo-4 signals were visualised by fluorescent microscopy (Olympus IX71, Japan). The readings from all fluo-4 positive cells in one measured field of view in each culture were included in the statistics, the fluorescence intensity of fluo-4 was normalised to the baseline intensity before stimulation. The measurements were repeated in 10 different cultures. Cultured astrocytes were also stained with specific marker sulforhodamine 101 (SR101) at 1:2000 dilution.

**Two-photon in vivo $Ca^{2+}$ imaging**. Adult FVB/N-Tg(GFAP-eGFP)14Mes/J transgenic mice (10–12 weeks old) were anesthetized with ketamine (80 mg/kg, i.p.) and xylazine (10 mg/kg, i.p.). Body temperature was monitored using a rectal probe, and the mice were maintained at 37 °C by a heating blanket. A custom made metal plate was glued to the skull with dental acrylic cement and a cranial window was prepared over the right hemisphere at 2.5 mm lateral and 2 mm posterior to bregma. The somato-sensory cortical cells were loaded with $Ca^{2+}$ indicator Rhod-2 AM (50 μM, 1 h). The transcranial window was superfused with artificial CSF. After a stable baseline recording was obtained, $Fe^{2+}$ (100 μM) or $Fe^{3+}$-TF (100 μM) was added for 1 min. Bandpass filters (Chroma) were 540 nm/40 nm for eGFP and 850 nm/70 nm for rhod-2 signals. Time-lapse images of astrocytic

$Ca^{2+}$ signalling were recorded every 5 s using FluoView with a custom-built two-photon laser-scanning setup (Nikon AR1, Japan)[84,85].

**Intracellular $Na^+$ measurements**. For monitoring intracellular ionised $Na^+$ ($[Na^+]_i$) in cultured astrocytes, primary cultured astrocytes were loaded with 10 μM of $Na^+$-sensitive indicator SBFI-AM for 30 min in serum-free medium, with subsequent 1 h of washout. SBFI was alternatively excited at 340 nm and 380 nm, and the emission was monitored at 500 nm. The SBFI signals were measured by fluorescent microscopy (Olympus IX71, Japan) and expressed as a ratio ($R = F_{340}/F_{380}$)[52].

**Immunofluorescence**. The brain tissue was fixed by immersion in 4% paraformaldehyde and cut into 100 μm slices[72,82]. The cultured cells were fixed with 100% methanol at −20 °C. Brain slices or cells were permeabilised by incubation for 1 h with donkey serum. Primary antibodies against DMT1 or TFR were used at a 1:100 dilution, against glial fibrillary acidic protein (GFAP) was used at 1:200 dilution. And nuclei were stained with marker 4′, 6′-diamidino-2-phenylindole (DAPI) at 1:1000 dilution. The primary antibodies were incubated overnight at 4 °C and then donkey anti-mouse or anti-rabbit Cy-2/3 conjugated secondary antibody for 2 h at room temperature. Images were captured using a confocal scanning microscope (DMi8, Leica, Wetzlar, Germany).

**ELISA assays**. Astrocytes were incubated at 37 °C in a fresh serum-free culture medium; after the treatment with $Fe^{2+}$/$Fe^{3+}$ or inhibitors, the astrocytes were collected and centrifuged at $10,000 \times g$ for 10 min to remove floating cells and/or cell debris at 4 °C. To assay the NKA activity, a commercial ELISA kit (abx255202; Abbexa, Cambridge, UK) was used and operated as the protocols, the sensitivity is 0.19 ng/mL, the optical density (OD) was measured at 450 nm and the OD value was normalized by the control group. To assay the $InsP_3$ concentration[86], the supernatant was collected and the concentration of $InsP_3$ assayed using a commercial ELISA kit (E-EL-0059c; Elabscience Biotechnology, Wuhan, China).

**Sorting neural cells through fluorescence activated cell sorter (FACS) and quantitative PCR (qPCR)**. To measure the mRNA for TFR and DMT1, astrocytes expressing fluorescent marker GFP (GFAP-GFP mice) and neurons expressing fluorescent marker YFP (Thy1-YFP mice) were used; we also extracted the cerebral hemispheres tissues from wild type mice. The cells from transgenic mice were used for specific sorting of astrocytes or neurons with FACS[85,87]. The RNA of the sorted cells and cerebral tissue was extracted by Trizol. Total RNA was reverse transcribed and PCR amplification was performed in a Robo-cycler thermocycler[81,84]. The relative quantity of transcripts was assessed using five-folds serial dilutions of RT product (200 ng). RNA quantity was normalised to glyceraldehyde 3-phosphate dehydrogenase (GAPDH) and values are expressed as the ratio TFR/GAPDH or DMT1/GAPDH.

**Statistics and reproducibility**. For statistical analysis, we used one-way analysis of variance (ANOVA) followed by a Tukey's or Dunnett's post hoc multiple comparison test for unequal replications using GraphPad Prism 5 software (GraphPad Software Inc., La Jolla, CA) and SPSS 24 software (International Business Machines Corp., NY, USA). One-way ANOVA for comparisons including more than two groups; unpaired two-tailed t-test for two-group comparisons. All statistical data in the text are presented as the mean ± SD, the value of significance was set at $p < 0.05$.

**Reporting summary**. Further information on research design is available in the Nature Research Reporting Summary linked to this article.

## Data availability
Source data underlying the main and supplementary figures are available in Supplementary Data. The data that support the findings of this study are available from the corresponding author Baoman Li upon reasonable request.

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

## Acknowledgements

This work was supported by the National Natural Science Foundation of China, B.L. [Grant number 81871852]; LiaoNing Revitalization Talents Program, B.L. [Grant number XLYC1807137], the Scientific Research Foundation for Returned Scholars of Education Ministry of China, B.L. [grant number 20151098], LiaoNing Thousands Talents Program, B.L. [Grant number 202078], "ChunHui" Program of China Education Ministry, B.L. [grant number 2020703], and the National Natural Science Foundation of China D.G. [Grant number 81671862] and [Grant number 81871529].

## Author contributions

A.V., D.G., M.X. and B.L. designed and supervised the study; M.X., WZ.G., S.L., G.W., BN.C., BJ.C., M.J. and WL.G. collected the data in vitro and analysed the relevant data; WZ.G., SS.L., C.D., X.Z., Y.Z., M.Z., D.Z. and X.L. performed the experiments in vivo and analysed the data; B.L. and A.V. wrote the manuscript. BN.C. and M.Z. collected the images of immunofluorescence, Figs. 1a, b and 2b, f.

## Competing interests

The authors declare no competing interests.
