## [Peer Review File · Communications Biology]

Reviewers' Comments:

Reviewer #1:

Remarks to the Author:

This manuscript addresses the significance of iron (Fe) ions on cytoplasmic free calcium ion concentration ($[Ca^{2+}]_i$) in astrocytes. Divalent metal ion transporter 1 (DMT1) and transferrin receptors (TFR) were particularly concentrated in the end-feet of cultured astrocytes. Micromolar Fe^{2+} or Fe^{3+} lead to Ca^{2+} signals within the astrocytes. Uptake of Fe^{2+} by DMT1 inhibited astroglial $Na^+-K^+-ATPase$ (NKA). Uptake of Fe^{3+} by TF-TFR stimulated phospholipase C to produce inositol 1,4,5-trisphosphate (InsP3) by triggering InsP3 receptor-mediated Ca^{2+} release from the endoplasmic reticulum. The iron-induced Ca^{2+} signals were suggested to promote release of arachidonic acid and prostaglandin E2 by activating cytosolic phospholipase A2 (cPLA2) and the NF- κ B signalling cascade.

The authors should take the following into advance:

Introduction: There is no solid proof stating that Fe^{2+} can exist in biological systems unless maintained in a prostetic core. Moreover, DMT1 was never really shown to be confined to cellular membranes of the brain (opposed to the gut). Please expand this description.

Result: It is not a surprise that Fe^{2+} is so reactive. It probably causes a direct chemical reaction with proteins of the cellular membranes that in turn become leaking.

Illustrations: The IHC shown in Fig 2B shows neuronal transferrin receptor staining. The DMT1 staining is mainly in the nucleus, probably of neurons. The authors need to demonstrate this much more convincingly. The literature does not attribute transferrin receptors to astrocytes, and these immunolabelings are not convincing.

The experiments on calcium are interesting but it can be speculated if the observations are mainly secondary phenomenons to general toxic events caused by the iron ions. The transferrin receptor mediated entry is probability the most physiological relevant, but the study does not really lead to conclusions on whether the iron can cause the observed changes as more that single events. Iron loading will downregulate transferrin receptors and hence reduce this form of iron uptake. Ferrous iron, as mentioned above, could mainly intoxicate the cells by direct interaction with the cellular membrane. Please explain.

Minor:

Introduction: correct "concentrated in endfeet of cerebral an hippocampal astrocytes"

Reviewer #2:

Remarks to the Author:

The manuscript titled "Iron transport and iron-induced Ca^{2+} signalling in astrocytes in vivo and in vitro" drew attention to astrocytic iron handling and to the intracellular pathways involved in calcium responses upon acute iron stimulation.

Although the topic of the manuscript is very interesting and opens to new prospective about the complex interplay between iron and calcium in the CNS, some experiments require further validation and some mechanisms postulated are not supported by strong experimental evidence. More in detail, the major criticisms are the following:

Major points

Results

Fe²⁺/Fe³⁺ trigger [Ca²⁺]_i increase in cortical astrocytes in vitro and in vivo.

The administration of Fe²⁺ induces fast calcium responses; I do not expect such a fast Fe²⁺ entry. The author should show the kinetics of Fe²⁺ entry by exploiting the capability of the ion to quench the fluorescence of some dyes, such as calcein or fura-2 (excited at a calcium insensitive wavelength, 360 nm). Moreover, did the authors verify whether Fe²⁺ entry can interfere with fluo-4 fluorescence?

Fig. 6: the authors propose a role of Dab2 protein in Tf internalization; however, they suggest a mechanism that bypasses the canonical known path (endocytosis, endosome acidification and Fe²⁺ release in the cytosol through DMT1) without supporting this new model.

In particular, at neutral cytosolic pH, is the Fe³⁺ able to detach from Tf? The author should better investigate the final nature of iron when they administer Tf-Fe³⁺ (again, tentatively by exploiting the characteristic of fura-2 vs calcium or by the use of specific iron chelators).

Fig. 7: the authors use BAPTA as selective Ca²⁺ chelator, without testing a possible same effects on Fe²⁺. In the presence of BAPTA, are they able to see a comparable fluorescence dye quenching (see first comment) than in the absence, therefore demonstrating that Fe²⁺ concentration is not affected by BAPTA? Without this control, the set of experiments with BAPTA doesn't make sense.

Discussion:

I'm concerned about the potentially neuroprotective effects the author discussed, based on their results; indeed, the acute administration of iron, although able to promote transient calcium responses and the consequent release of protective molecules (in the discussion the authors should add some references about the neuroprotective role of AA and PGE₂), does not reproduce what happens chronically, during neurodegenerative disorders. Despite the experimental time window, a chronic treatment of astrocytes with iron (Fe³⁺) would help to figure out how long the release of AA or PGE₂ lasts and thus which could be the protective relevance. Every stimulus able to promote transient calcium increase can activate the same intracellular cascade. Moreover, the simple release of PGE₂ is not sufficient to speculate about the role of astrocytes in protecting brain against iron overload (last paragraph of the Discussion); the activation of astrocytes (mediated by an excess of iron or, more in general, by neurodegenerative conditions) can occur also towards a neurotoxic phenotype.

Minor points:

The manuscript requires a thorough revision of the spelling (a lot of misprints are present).

Fig.1 is very confusing, with the panels that are not clearly defined; this is further worsened by the fact that, in the Results section, the panel B is described after panel C. The logical sequence in the text should be maintained in the figure

DMT1 and TFR mediate Fe²⁺ and Fe³⁺ uptake

-The first sentence "Fe²⁺ uptake is mediated by plasmalemmal transporter DMT1," is too strong; other can account for Fe²⁺ entry;

-Fig 2C: I'm surprised about the expression of DMT1 as well as TFR mRNAs that, in cortical tissues, appear very similar to those extracted from neurons, as if astrocytes, present at elevated % in the brain, do not give any relevant contribute

Fig. 2F: What about the mechanism of DMT1 redistribution? Is it calcium-mediated?

Fig. 5: the Authors did not specify how long, after the 5 min treatment with Fe²⁺, they lysate the astrocytes for WB analysis.

Discussion: Aceruplminemia and Neuroferritinopathy belong to NBIA

Reviewer #3:

Remarks to the Author:

Maosheng Xia et al., 2020. Communications Biology

Title: Iron transport and iron-induced Ca²⁺ signaling in astrocytes in vivo and in vitro.

In the present study, the authors analyzed intracellular calcium changes in astrocytes induced by iron. The investigators performed mostly in vitro studies but they also conducted some in vivo experiments in cortical astrocytes. The data suggest that iron produce an increase in the intracellular calcium concentration of astrocytes by acting through two distinct mechanisms. Uptake of ferrus iron by DMT1 inhibited astroglial Na⁺-K⁺-ATPase, which led to an elevation in cytoplasmic Na⁺ concentration, thus reversing the Na⁺/Ca²⁺ exchanger which generates Ca²⁺ uptake. In contrast, ferric iron is incorporated in astrocytes via the transferrin (Tf) cycle. This activates InsP3 production, which induce the release of calcium from the endoplasmic reticulum. Finally, they demonstrate that iron-induced calcium signals promote the release of arachidonic acid and prostaglandin E2 in astrocytes.

In summary, using a combination of specific pharmacological inhibitors, siRNAs and calcium imaging, the investigators described two new mechanisms of iron-induced astrocytic calcium signaling. Overall, this is an interesting and well-conducted study and the quality of the data is good. During development as well as under pathological situations astrocytes are key player in brain iron metabolism. Thus, the conclusions presented by the authors are relevant and important to advance understanding the role of astrocytes in neurodevelopment.

I would like to recommend some additional experiments/analysis to complete and to increase the relevance/impact of the study.

Main Points:

- Figure 1. The size as well as the resolution of the pictures showed in panels A) and B) should be increased. It will be also important to specify which area of the cortex is being analyzed in the in vivo experiments.
- The purity of the primary astrocyte cultures should be more carefully evaluated with cell markers other than GFAP. Additionally, showing some astrocytes markers in culture experiments will be essential to demonstrate that the calcium recordings are been performed in astrocytes.
- In vitro as well as in vivo astrocytes are usually inter-connected by gap junctions. Does iron stimulation induces calcium waves in these cells?
- Figure 2. The plasma membrane localization of DMT1 and transferrin receptor in astrocytes should be more carefully analyzed in vitro as well as in vivo by measuring fluorescent intensity in specific cell compartments. Additionally, it will be important to show high magnification pictures with good resolution to visualize the cellular distribution of these proteins more clearly. It will be also interesting to study changes in the expression and/or the intracellular location of DMT1 and Tf receptor between cultured cell and astrocytes in tissue preparations.
- The illustration of the transferrin cycle is incorrect. The Tf receptor-holo-Tf complex undergoes endocytosis through clathrin pit formation. The endosome then acidifies, and the endosomal metalloredutase reduces Fe³⁺ to Fe²⁺, allowing iron, now released from Tf, to be transported into the cytosol by DMT1. Thus, DMT1 should be also included in the Tf cycle.

- Figure 3. The calcium imaging experiments presented in this figure should be accompanied with graph bars and statistical analysis. It is not clear if the calcium traces are showing the response of one single astrocyte or in fact are the average or several cells. It will be also important to quantify the percentage of responding cell for each pharmacological treatment.

- Figure 6 and 8. The complete transferrin cycle should be included in the proposed mechanisms.

Minor Points:

- A significant portion of the Discussion section is devoted to recapitulate the results, it will be important to discuss the physiological relevance of the conclusions and how these new data improve our understanding of the role of astrocytes in brain iron metabolism.

- The entire manuscript should be revised for typographical and grammar mistakes.

Responses to COMMSBIO-20-2725-T

Reviewer #1:

This manuscript addresses the significance of iron (Fe) ions on cytoplasmic free calcium ion concentration ($[Ca^{2+}]_i$) in astrocytes. Divalent metal ion transporter 1 (DMT1) and transferrin receptors (TFR) were particularly concentrated in the end-feet of cultured astrocytes. Micromolar Fe^{2+} or Fe^{3+} lead to Ca^{2+} signals within the astrocytes. Uptake of Fe^{2+} by DMT1 inhibited astroglial $Na^+-K^+-ATPase$ (NKA). Uptake of Fe^{3+} by TF-TFR stimulated phospholipase C to produce inositol 1,4,5-trisphosphate (InsP3) by triggering InsP3 receptor-mediated Ca^{2+} release from the endoplasmic reticulum. The iron-induced Ca^{2+} signals were suggested to promote release of arachidonic acid and prostaglandin E2 by activating cytosolic phospholipase A2 (cPLA2) and the NF- κ B signalling cascade.

The authors should take the following into advance:

Introduction: There is no solid proof stating that Fe^{2+} can exist in biological systems unless maintained in a prostetic core. Moreover, DMT1 was never really shown to be confined to cellular membranes of the brain (opposed to the gut). Please expand this description.

Our reply: DMT1 has been identified in various cellular elements in the brain. For example, DMT1 expression has been found (i) in the whole brain (Wu et al., 2021; Zhang et al., 2020) (ii) in neurones (Wu et al., 2020) and (iii) in glia at mRNA levels and protein levels (Aral et al., 2020).

For the clear description, we added the relevant statements about DMT1 in second paragraph of “Introduction”.

Wu Q, Hao Q, Li H, Wang B, Wang P, Jin X, Yu P, Gao G, Chang YZ. Brain iron deficiency and affected contextual fear memory in mice with conditional Ferroportin1 ablation in the brain. *FASEB J*. 2021 Feb;35(2):e21174. doi: 10.1096/fj.202000167RR. Epub 2020 Nov 16. PMID: 33200454.

Zhang L, Xiao H, Yu X, Deng Y. Minocycline attenuates neurological impairment and regulates iron metabolism in a rat model of traumatic brain injury. *Arch Biochem Biophys*. 2020 Mar 30;682:108302. doi: 10.1016/j.abb.2020.108302. Epub 2020 Feb 10. PMID: 32057758.

Wu J, Yang JJ, Cao Y, Li H, Zhao H, Yang S, Li K. Iron overload contributes to general anaesthesia-induced neurotoxicity and cognitive deficits. *J Neuroinflammation*. 2020 Apr 11;17(1):110. doi: 10.1186/s12974-020-01777-6. PMID: 32276637; PMCID: PMC7149901.

Aral LA, Ergün MA, Engin AB, Börcek AÖ, Belen HB. Iron homeostasis is altered in response to hypoxia and hypothermic preconditioning in brain glial cells. *Turk J Med Sci*. 2020 Dec 17;50(8):2005-2016. doi: 10.3906/sag-2003-41. PMID: 32682355; PMCID: PMC7775693.

Result: It is not a surprise that Fe^{2+} is so reactive. It probably causes a direct chemical reaction with proteins of the cellular membranes that in turn become leaking.

Our reply: We did not observe any reactivity of Fe (and we are not even certain which kind of reactivity can be considered), neither we observed any unspecific action of Fe linked, for example, to an increased membrane permeability. To the contrary, we found that Fe²⁺ and Fe³⁺ activate two distinct molecular cascades which mediate Fe entry and Fe-induced Ca²⁺ signalling. These different cascades are reflected by the different kinetics of [Ca²⁺]_i responses induced by two iron forms and by the fact that Fe²⁺-dependent Ca²⁺ signalling required extracellular Ca²⁺, whereas Fe³⁺ was independent from it. Operation of each of these molecular cascades has been tested and proved pharmacologically and by selective genes silencing. We found no evidence for leaking cellular membranes; our cells were filled with low-molecular weight Ca or Na probes, which did not display any signs of leak. To prove the specific effects of Fe²⁺ or Fe³⁺ on the astrocytic Ca²⁺ signalling, the expression of DMT1 or TFR in astrocytes were RNA interfered with siRNA duplex, respectively (Fig. 2E). To further prove the specific effect of Fe²⁺ on NCX, we used NCX selective inhibitor, KB-RT943 (Fig. 3D); similarly, the specific effect of Fe³⁺ on ER Ca²⁺ release was further corroborated by InsP₃R inhibitor, XeC and by PLC blocker which both potently suppressed Fe³⁺-induced Ca²⁺ signals (Fig. 3F).

Illustrations: The IHC shown in Fig 2B shows neuronal transferrin receptor staining. The DMT1 staining is mainly in the nucleus, probably of neurons. The authors need to demonstrate this much more convincingly. The literature does not attribute transferrin receptors to astrocytes, and these immunolabelings are not convincing.

Our reply: To reveal cellular localisation of both TFR and DMT1 we used double staining with antibodies against DMT1 or TFR and GFAP; in the revised version we provide high-resolution images (Fig. 2B, F) which, we believe, clearly demonstrate co-localisation of both proteins with GFAP positive profiles (i.e. their expression in astrocytes) in the brain tissue. Immunocytochemistry *in vitro* was done in a similar fashion and again we now present high resolution images which clearly demonstrate astrocytic expression of both DMT1 and TFR (Fig. 2 B). Of note, our cultures have >95% purity as judged by GFAP staining; these cultures realistically do not contain neurones.

We also found that incubation with Fe²⁺ induces rapid redistribution of DMT1 from nucleus into the cytoplasm. To further corroborate this, we extracted the proteins of astrocytic nucleus and cytoplasm after the administration of ferrous for 5 mins, then the levels of DMT1 were separately measured by western-blotting; these new results are added as Fig. 2G. We also extract RNA from FACS sorted astrocytes from GFAP-eGFP transgenic mice and demonstrated mRNA expression for both DMT1

and TRF specifically in purified acutely isolated astrocytes (Fig. 2C). We found that mRNA levels for both proteins did not differ significantly between neurons and astrocytes. We addressed all these matters in the discussion in the revised version.

About transferrin; indeed there are controversies (which we hope our paper solves) about the expression of TFR in astrocytes *in vivo*, nonetheless expression in primary cultures have been frequently reported (Neuropeptides, 48(3):161-166, 2014; Biochim Biophys Acta, 1832(8):1326-1333, 2013).

The experiments on calcium are interesting but it can be speculated if the observations are mainly secondary phenomenons to general toxic events caused by the iron ions. The transferrin receptor mediated entry is probability the most physiological relevant, but the study does not really lead to conclusions on whether the iron can cause the observed changes as more that single events. Iron loading will downregulate transferrin receptors and hence reduce this form of iron uptake. Ferrous iron, as mentioned above, could mainly intoxicate the cells by direct interaction with the cellular membrane. Please explain.

Our reply: Effects of Fe^{2+} and Fe^{3+} on the intracellular Ca^{2+} are highly specific and distinct, they cannot be explained by deterioration of the cellular membrane or by any other secondary phenomena. Intracellular Ca^{2+} signals triggered by Fe^{2+} and Fe^{3+} have distinct properties with distinct pharmacology and distinct dependence on silencing specific genes. Both pharmacology and gene silencing show the involvement of DMT1, Na/K pump and NCXs in the Fe^{2+} -induced Ca^{2+} signalling and intracellular stores in Fe^{3+} -induced Ca^{2+} signalling.

Chronic exposure to iron, as we have shown in our previous work, up-regulates TFR expression in neural cells (Neurosci Bull, 36(12):1542-1546, 2020).

Minor:

Introduction: correct “concentrated in endfeet of cerebral an hippocampal astrocytes”

Our reply: Thanks, we corrected it.

Reviewer #2:

The manuscript titled “Iron transport and iron-induced Ca²⁺ signalling in astrocytes in vivo and in vitro” drew attention to astrocytic iron handling and to the intracellular pathways involved in calcium responses upon acute iron stimulation.

Although the topic of the manuscript is very interesting and opens to new prospective about the complex interplay between iron and calcium in the CNS, some experiments require further validation and some mechanisms postulated are not supported by strong experimental evidence.

More in detail, the major criticisms are the following:

Our reply: Thanks for your comments, according to your suggestions we performed new experiments, provided higher resolution images and made further discussion about the mechanisms concerned.

Major points

Results

Fe²⁺/Fe³⁺ trigger [Ca²⁺]_i increase in cortical astrocytes in vitro and in vivo.

The administration of Fe²⁺ induces fast calcium responses; I do not expect such a fast Fe²⁺ entry. The author should show the kinetics of Fe²⁺ entry by exploiting the capability of the ion to quench the fluorescence of same dyes, such as calcein or fura-2 (excited at a calcium insensitive wavelength, 360 nm). Moreover, did the authors verify whether Fe²⁺ entry can interfere with fluo-4 fluorescence?

Our reply: We have monitored Fe entry using quenching of Fura-2 (in Supplementary Fig. 1); indeed Fe entry is not very fast and yet it is already significant within first 10s after the beginning of application. The quenching reaches saturation in about 120 s after the beginning of application indicating the existence of specific and saturatable transport pathways. The Ca²⁺ signals we record from astrocytes are similarly not very fast: the peak of [Ca²⁺]_i response is reached at ~ 15-20 s after the start of Fe application which is consistent with quenching data. We did verify the absence of interactions between both forms of ionized iron with neither fluo-4 nor Rhod-2 in a cell free experiment.

Fig. 6: the authors propose a role of Dab2 protein in Tf internalization; however, they suggest a mechanism that bypasses the canonical known path (endocytosis, endosome acidification and Fe²⁺ release in the cytosol through DMT1) without supporting this new model.

In particular, at neutral cytosolic pH, is the Fe³⁺ able to detach from TF? The author should better investigate the final nature of iron when they administer Tf-Fe³⁺ (again, tentatively by exploiting the characteristic of fura-2 vs calcium or by the use of specific iron chelators).

Our reply: This is a good question. According to our results that there is no direct interaction between Fe²⁺-induced Ca²⁺ signalling (which involves DMT1, NKA and reversed NCX) and by Fe³⁺-induced Ca²⁺ signalling (which depends on InsP₃ receptors

and can be completely blocked by XeC (inhibitor of InsP₃ receptor) (Fig. 3E) and U-73122 (inhibitor of PLC) (Fig. 6D). The action of Fe³⁺ is also completely suppressed by TFR siRNA (Fig. 2E). As to the role of DMT1 in Fe³⁺ release from endosomes - we performed additional experiments with DMT1 silencing; it turns out that inhibition of DMT1 does not affect Fe³⁺-induced Ca²⁺ signalling (Supplementary Fig. 2), thus questioning the role of DMT1 in this pathway. It seems that Fe³⁺ is released from endosomes by another pathway. We added endosomal pathways into our schemes and discussed possible mechanism of Fe³⁺ release. Please see the fourth paragraph in the Discussion of “Mechanisms of iron induced Ca²⁺ signalling”. During the internalization of complex, Fe³⁺-TF binding TFR requires the pH around 7.4, then within the endosome the complex is acidified through ATP-dependent H⁺ pumps until pH 5.6, Fe³⁺ could be released from TF. Because this process is very complex, many potential targets are involved in the stimulation of PLC, we will do the further research in our future work.

Fig. 7: the authors use BAPTA as selective Ca²⁺ chelator, without testing a possible same effects on Fe²⁺. In the presence of BAPTA, are they able to see a comparable fluorescence dye quenching (see first comment) than in the absence, therefore demonstrating that Fe²⁺ concentration is not affected by BAPTA? Without this control, the set of experiments with BAPTA doesn't make sense.

Our reply: We agree entirely and we therefore removed all BAPTA experiments; we also realised that our data on cPLA₂, AA and PGE₂ do not fit into the present story and hence we removed them (and corresponding figure) altogether.

Discussion:

I'm concerned about the potentially neuroprotective effects the author discussed, based on their results; indeed, the acute administration of iron, although able to promote transient calcium responses and the consequent release of protective molecules (in the discussion the authors should add some references about the neuroprotective role of AA and PGE₂), does not reproduce what happens chronically, during neurodegenerative disorders. Despite the experimental time window, a chronic treatment of astrocytes with iron (Fe³⁺) would help to figure out how long the release of AA or PGE₂ lasts and thus which could be the protective relevance. Every stimulus able to promote transient calcium increase can activate the same intracellular cascade. Moreover, the simple release of PGE₂ is not sufficient to speculate about the role of astrocytes in protecting brain against iron overload (last paragraph of the Discussion); the activation of astrocytes (mediated by an excess of iron or, more ingeneral, by neurodegenerative conditions) can occur also towards a neurotoxic phenotype.

Our reply: We agree; the AA and PGE cascades need more investigation in the cortex of neuroprotection and therefore we removed this set of data and modified discussion accordingly.

Minor points:

The manuscript requires a thorough revision of the spelling (a lot of misprints are present).

Our reply: we performed careful proofread, hoping to eliminate all errors.

Fig.1 is very confusing, with the panels that are not clearly defined; this is further worsened by the fact that, in the Results section, the panel B is described after panel C. The logical sequence in the text should be maintained in the figure

Our reply: We have modified the text to follow logical sequence.

DMT1 and TFR mediate Fe²⁺ and Fe³⁺ uptake

-The first sentence "Fe²⁺ uptake is mediated by plasmalemmal transporter DMT1," is too strong; other can account for Fe²⁺ entry;

Our reply: Thanks for the suggestion, we have modified this sentence.

Fig 2C: I'm surprised about the expression of DMT1 as well as TFR mRNAs that, in cortical tissues, appear very similar to those extracted from neurons, as if astrocytes, present at elevated % in the brain, do not give any relevant contribute

Our reply: The common notion that astrocytes outnumber neurons in humans ("present at elevated % in the brain") is incorrect (see Verkhratsky, Butt *Neuroglia*, v. 1, p. 188 - 192); in the rodent brain the ratio astrocytes to neurones is ~0.2 (Keller e.a. *Front Neuroanat* 12:83).

Fig. 2F: What about the mechanism of DMT1 redistribution? Is it calcium-mediated?

Our reply: To further prove the redistribution of DMT1 from nucleus into the cytoplasm, we extracted the proteins of astrocytic nucleus and cytoplasm after the administration of ferrous for 5 mins, then the levels of DMT1 were separately measured by western-blotting, please see new Fig. 2G. We do not now, however, what is the underlying mechanism, elucidation of which required a dedicated project and is, as we feel, beyond the scope of the present paper.

Fig. 5: the Authors did not specify how long, after the 5 min treatment with Fe²⁺, they lysate the astrocytes for WB analysis. At once, after 5 min, we used the lysate to stop action, so the release AA or PGE2 should be in 5 mins.

Our reply: We collected the protein using lysis buffer after the treatment with iron for 5 mins.

Discussion: Aceruplminemia and Neuroferritinopathy belong to NBIA

Our reply: We added the relevant discussion on these two diseases to the paragraph "Do astrocytes protect the brain against iron overload?".

Reviewer #3:

In the present study, the authors analyzed intracellular calcium changes in astrocytes induced by iron. The investigators performed mostly in vitro studies but they also conducted some in vivo experiments in cortical astrocytes. The data suggest that iron produce an increase in the intracellular calcium concentration of astrocytes by acting through two distinct mechanisms. Uptake of ferrus iron by DMT1 inhibited astroglial Na⁺-K⁺-ATPase, which led to an elevation in cytoplasmic Na⁺ concentration, thus reversing the Na⁺/Ca²⁺ exchanger which generates Ca²⁺ uptake. In contrast, ferric iron is incorporated in astrocytes via the transferrin (Tf) cycle. This activates InsP3 production, which induce the release of calcium from the endoplasmic reticulum. Finally, they demonstrate that iron-induced calcium signals promote the release of arachidonic acid and prostaglandin E2 in astrocytes.

In summary, using a combination of specific pharmacological inhibitors, siRNAs and calcium imaging, the investigators described two new mechanisms of iron-induced astrocytic calcium signaling. Overall, this is an interesting and well-conducted study and the quality of the data is good. During development as well as under pathological situations astrocytes are key player in brain iron metabolism. Thus, the conclusions presented by the authors are relevant and important to advance understanding the role of astrocytes in neurodevelopment.

I would like to recommend some additional experiments/analysis to complete and to increase the relevance/impact of the study.

Main Points:

Figure 1. The size as well as the resolution of the pictures showed in panels A) and B) should be increased. It will be also important to specify which area of the cortex is being analyzed in the in vivo experiments.

Our reply: We present new high resolution images in Figs 1 and 2; we also added a schematic indicating the recording area (somato-sensory cortex).

The purity of the primary astrocyte cultures should be more carefully evaluated with cell markers other than GFAP. Additionally, showing some astrocytes markers in culture experiments will be essential to demonstrate that the calcium recordings are been performed in astrocytes.

Our reply: We assessed the purity of cultures with both GFAP and GLT1, please see Fig. 1A. The purity of astrocytes according to GFAP staining is more than 95%, as we have shown many times in our previous papers (see e.g. Neurochem Int, 134, 104689, 2020). To further corroborate astrocyte-specificity of recordings we co-stained our cultures with sulforhodamine 101, a popular vital astrocyte marker (Sci Rep, 6:30433,

2016).

In vitro as well as in vivo astrocytes are usually inter-connected by gap junctions. Does iron stimulation induces calcium waves in these cells?

Our reply: In this study, we mainly focus on elucidating the mechanism of iron-induced Ca^{2+} signalling; hence we have not investigated propagating Ca^{2+} waves. This however is a valid suggestion and we shall explore this in our future work.

Figure 2. The plasma membrane localization of DMT1 and transferrin receptor in astrocytes should be more carefully analyzed in vitro as well as in vivo by measuring fluorescent intensity in specific cell compartments. Additionally, it will be important to show high magnification pictures with good resolution to visualize the cellular distribution of these proteins more clearly. It will be also interesting to study changes in the expression and/or the intracellular location of DMT1 and Tf receptor between cultured cell and astrocytes in tissue preparations.

Our reply: In Fig. 2, we now added high resolution images for panels B and F. To further corroborate the redistribution of DMT1 induced by Fe^{2+} from nucleus to cytoplasm, we added the measurements of DMT1 separately in nucleus and cytoplasm, please see new Fig. 2G. Concerning expression in the tissue - these are future plans of course; in our recent papers, we report that chronic iron over-load increases the expression of DMT1 in glial cells (Xia et al., 2021) and up-regulates TFR in cortex (Liang et al., 2020). We have added this to discussion in “Iron transport in astrocytes is mediated by DMT1 and TFR”. Finally, as the sensitivity of detection in two-photon is limited, we could not monitor the redistribution of DMT1 from nucleus to cytoplasm in the in vivo mice.

The illustration of the transferrin cycle is incorrect. The Tf receptor-holo-Tf complex undergoes endocytosis through clathrin pit formation. The endosome then acidifies, and the endosomal metallo-reductase reduces Fe^{3+} to Fe^{2+} , allowing iron, now released from Tf, to be transported into the cytosol by DMT1. Thus, DMT1 should be also included in the Tf cycle.

Our reply: As to the role of DMT1 in Fe^{3+} release from endosomes - we performed additional experiments with DMT1 silencing; it turns out that inhibition of DMT1 does not affect Fe^{3+} -induced Ca^{2+} signalling (Supplementary Fig. 2), thus questioning the role of DMT1 in this pathway. It seems that Fe^{3+} is released from endosomes by another pathway. We added endosomal pathways into our schemes (Figs 6 and 8) and discussed possible mechanism of Fe^{3+} release. Please see the fourth paragraph in the Discussion of “Mechanisms of iron induced Ca^{2+} signalling”. It remains however

unclear whether the stimulation of PLC by the internalized of TFR-TF-Fe³⁺ requires the release of Fe³⁺ from TF/endosome, and we are thinking about new experimental project to address this issue.,

Figure 3. The calcium imaging experiments presented in this figure should be accompanied with graph bars and statistical analysis. It is not clear if the calcium traces are showing the response of one single astrocyte or in fact are the average or several cells. It will be also important to quantify the percentage of responding cell for each pharmacological treatment.

Our reply: In vitro, we traced all fluo-4 positive cells for one experimental protocol in each culture, there were around 5-7 cells in one measured field of view, and the fluorescence intensity of fluo-4 was normalised to the baseline intensity before stimulation; all experiments were repeated in 10 different cultures; we added this information to the paper. All Ca²⁺ traces are averages of the intensity of fluo4 ± SD, and n=10. In vivo, we firstly used GFAP-eGFP to identify individual astrocyte and the images Rhod-2 the same measurement was repeated in 10 different mice.

Figure 6 and 8. The complete transferrin cycle should be included in the proposed mechanisms.

Our reply: We modified the diagram in Fig. 6G and (new) Fig. 7 as suggested.

Minor Points:

A significant portion of the Discussion section is devoted to recapitulate the results, it will be important to discuss the physiological relevance of the conclusions and how these new data improve our understanding of the role of astrocytes in brain iron metabolism.

Our reply: We added the relevant discussion about physiological meaning of iron metabolism, please see the red words in “Iron transport in astrocytes is mediated by DMT1 and TFR” and “Do astrocytes protect the brain against iron overload?”.

The entire manuscript should be revised for typographical and grammar mistakes.

Our reply: Thanks, we have tried our best to modify all written errors.

Reviewers' Comments:

Reviewer #1:

Remarks to the Author:

Accept

Reviewer #2:

Remarks to the Author:

The manuscript titled "Iron transport and iron-induced Ca²⁺ signalling in astrocytes in vivo and in vitro" addresses an interesting topic that concerns the complex interplay between iron and calcium, two key ions for physiological brain functions, but also involved in neuronal disorders. The authors investigated iron handling in astrocytes and the mechanisms involved in calcium elevation upon acute iron stimulation.

In the revised version, part of the previous manuscript has been deleted (fig. 7 and all the neuroprotective implications proposed in the Discussion section), but the results were based on wrong experimental design, accordingly to my indication; therefore, despite the reduced number of the data shown now, I think that the manuscript benefited from this.

No major changes are required for the publication of the manuscript.

Reviewer #3:

Remarks to the Author:

Maosheng Xia et al., 2020. Communications Biology

Title: Iron transport and iron-induced Ca²⁺ signaling in astrocytes in vivo and in vitro.

This revised manuscript is improved following the recommendations of the reviewers. In the current version of the manuscript new in vitro and in vivo experiments were included to show the effect of iron on cortical astrocytes. As a consequence, the present manuscript clearly demonstrates that iron produce an increase in the intracellular calcium concentration of astrocytes by acting through DMT1 and the transferrin receptor on the plasma membrane.

Responses to COMMSBIO-20-2725-T

Thanks for all reviewers' comments, which promote our study.

Reviewer #1 (Remarks to the Author):

Accept

Reviewer #2 (Remarks to the Author):

The manuscript titled “Iron transport and iron-induced Ca²⁺ signalling in astrocytes in vivo and in vitro” addresses an interesting topic that concerns the complex interplay between iron and calcium, two key ions for physiological brain functions, but also involved in neuronal disorders. The authors investigated iron handling in astrocytes and the mechanisms involved in calcium elevation upon acute iron stimulation.

In the revised version, part of the previous manuscript has been deleted (fig. 7 and all the neuroprotective implications proposed in the Discussion section), but the results were based on wrong experimental design, accordingly to my indication; therefore, despite the reduced number of the data shown now, I think that the manuscript benefited from this.

No major changes are required for the publication of the manuscript.

Reviewer #3 (Remarks to the Author):

Maosheng Xia et al., 2020. Communications Biology

Title: Iron transport and iron-induced Ca²⁺ signaling in astrocytes in vivo and in vitro.

This revised manuscript is improved following the recommendations of the reviewers. In the current version of the manuscript new in vitro and in vivo experiments were included to show the effect of iron on cortical astrocytes. As a consequence, the present manuscript clearly demonstrates that iron produce an increase in the intracellular calcium concentration of astrocytes by acting through DMT1 and the transferrin receptor on the plasma membrane.